**Colony formation in *Phaeocystis antarctica*: connecting molecular**
**mechanisms with iron biogeochemistry**
Sara J. Bender[a,b], Dawn M. Moran[a], Matthew R. McIlvin[a], Hong Zheng[c], John P. McCrow[c],
Jonathan Badger[c,f], Giacomo R. DiTullio[e], Andrew E. Allen[c,d], Mak A. Saito[a,*]
[a]Marine Chemistry and Geochemistry Department, Woods Hole Oceanographic Institution,
Woods Hole, Massachusetts 02543 USA
[b]Current address: Gordon and Betty Moore Foundation, Palo Alto, California 94304 USA
[c]Microbial and Environmental Genomics, J. Craig Venter Institute, La Jolla, California 92037
USA
[d]Integrative Oceanography Division, Scripps Institution of Oceanography, UC San Diego, La
Jolla, California 92037 USA
[e]College of Charleston, Charleston South Carolina 29412, USA
[f]Current address: Center for Cancer Research, Bethesda, Maryland 20892, USA
[*]Correspondence to M. Saito (msaito@whoi.edu)
*Accepted at Biogeosciences*
*July 31, 2018*
*Minor Technical Corrections*

**Abstract.**

*Phaeocystis antarctica* is an important phytoplankter of the Ross Sea where it dominates the early season bloom after sea ice retreat and is a major contributor to carbon export. The factors that influence *Phaeocystis* colony formation and the resultant Ross Sea bloom initiation have been of great scientific interest, yet there is little known about the underlying mechanisms responsible for these phenomena. Here, we present laboratory and field studies on *Phaeocystis antarctica* grown under multiple iron conditions using a coupled proteomic and transcriptomic approach. *P. antarctica* had a lower iron limitation threshold than a Ross Sea diatom *Chaetoceros* sp., and at increased iron nutrition (>120 pM Fe') a shift from flagellate cells to a majority of colonial cells in *P. antarctica* was observed, implying a role for iron as a trigger for colony formation. Proteome analysis revealed an extensive and coordinated shift in proteome structure linked to iron availability and life cycle transitions with 327 and 436 of proteins measured were significantly different between low and high iron in strains 1871 and 1374, respectively. The enzymes flavodoxin and plastocyanin that can functionally replace iron metalloenzymes were observed at low iron treatments consistent with cellular iron sparing strategies, with plastocyanin having a larger dynamic range. The numerous isoforms of the putative iron-starvation induced protein ISIP group (ISIP2A and ISIP3) had abundance patterns coincided with that of either low or high iron (and coincident flagellate or the colonial cell types in strain 1871), implying that there may be specific iron acquisition systems for each life cycle type. The proteome analysis also revealed numerous structural proteins associated with each cell type: within flagellate cells actin and tubulin from flagella and haptonema structures as well as a suite of calcium-binding proteins with EF domains were observed. In the colony-dominated samples a variety of structural proteins were observed that are also often found in multicellular organisms including spondins, lectins, fibrillins,

and glycoproteins with von Willebrand domains. A large number of proteins of unknown function
were identified that became abundant at either high and low iron availability. These results were
compared to the first metaproteomic analysis of a Ross Sea *Phaeocystis* bloom to connect the
mechanistic information to the *in situ* ecology and biogeochemistry. Proteins associated with both
flagellate and colonial cells were observed in the bloom sample consistent with the need for both
cell types within a growing bloom. Bacterial iron storage and $B_{12}$ biosynthesis proteins were also
observed consistent with chemical synergies within the colony microbiome to cope with the
biogeochemical conditions. Together these responses reveal a complex, highly coordinated effort
by *P. antarctica* to regulate its phenotype at the molecular level in response to iron and provide a
window into the biology, ecology, and biogeochemistry of this group.
**1.   Introduction**
The genus *Phaeocystis* is a cosmopolitan marine phytoplankton group that plays a key role in
global carbon and sulfur cycles (Hamm et al., 1999; Matrai et al., 1995; Rousseau et al., 2007;
Schoemann et al., 2005; Smith et al., 1991; Solomon et al., 2003; Thingstad and Billen, 1994;
Verity et al., 2007). Because of their large cell concentrations during bloom formation, *Phaeocystis*
have a significant impact on the ocean biogeochemistry through carbon fixation (Arrigo et al.,
1999; Hamm et al., 1999; Matrai et al., 1995; Rousseau et al., 2007; Schoemann et al., 2005; Smith
et al., 1991; Solomon et al., 2003; Thingstad and Billen, 1994; Verity et al., 2007), the release of
large concentrations of organic carbon upon grazing/viral lysis (Alderkamp et al., 2007; Hamm et
al., 1999; Lagerheim, 1896; Verity et al., 2007), and export as aggregates out of the photic zone
(DiTullio et al., 2000). Through the production of dimethylsulfide (DMS), they also directly
connect ocean and atmospheric processes and carbon and sulfur cycling (Smith et al., 2003).

Some *Phaeocystis* species, including *Phaeocystis antarctica*, undergo multiple

morphotypes and can occur as flagellated single-cells or in gelatinous colonies consisting of
thousands of non-motile cells (Fig. 1). Microscopic and chemical analyses have found that
*Phaeocystis* colonies are filled with a mucilaginous matrix surrounded by a thin, but strong,
hydrophobic skin (Hamm, 2000; Hamm et al., 1999). Once formed, cells typically associate with
this outer layer of the colony (Smith et al., 2003). Colony formation involves the exudation of
(muco)polysaccharides and carbohydrate-rich dissolved organic matter, as well as amino sugars
and amino acids; it is estimated that approximately 50 – 80% of *Phaeocystis* carbon is allocated to
this extracellular matrix (Hamm et al., 1999; Matrai et al., 1995; Rousseau et al., 2007; Solomon
et al., 2003; Thingstad and Billen, 1994).  Thus, not only does the colony increase the size of
*Phaeocystis* by several orders of magnitude, but the extracellular matrix material also constitutes
the majority of measured algal (carbon) biomass (Rousseau et al., 1990). The colonial form of
*Phaeocystis* has been suggested as a defense mechanism against grazers (Hamm et al., 1999), a
means to sequester micronutrients such as iron and manganese (Noble et al., 2013; Schoemann et
al., 2001), as a means of protection from pathogens (Hamm, 2000; Jacobsen et al., 2007), and as a
microbiome vitamin $B_{12}$ source (Bertrand et al., 2007). Colony formation of *Phaeocystis* species,
including *P. antarctica* and *P. globosa*, has been linked to numerous physiological triggers
including the synergistic effects of iron and irradiance (Feng et al., 2010), grazer-induced chemical
cues (Long et al., 2007), phosphate concentrations (Riegman et al., 1992), and the presence of
different nitrogen species (Riegman and van Boekel, 1996; Smith et al., 2003).

The Ross Sea is one of the most productive regions of the Southern Ocean (Arrigo et al.,

1999; 1998; Feng et al., 2010; Garcia et al., 2009; Sedwick and DiTullio, 1997), and the latter is
an important contributor to the cycling of carbon in the oceans (Lovenduski et al., 2008; Sarmiento
et al., 1998). In the early spring when the sea ice retreats and polynyas form, phytoplankton blooms
and regional phytoplankton productivity are fed by the residual winter iron inventory and perhaps
iron-rich sea ice melt (Noble et al., 2013; Sedwick and DiTullio, 1997); blooms have also been
linked to changes in irradiance and mixed layer depth (Arrigo et al., 1999; Coale et al., 2003;
Martin et al., 1990; Sedwick and DiTullio, 1997; Sedwick et al., 2000). In the Ross Sea Polynya
(RSP), *P. antarctica* colonial cells form almost mono-specific blooms until the austral
summer season begins, comprising > 98% of cell abundance at the peak of the bloom (Smith et
al., 2003). Although diatom abundance dominates in the summer, the RSP typically harbors
the co-existence of flagellated single cells of *P. antarctica* along with diatoms (Garrison et al.,
2003). During blooms *P. antarctica* can draw down more than twice as much carbon relative to
phosphate as diatoms and contribute to rapid carbon export, leaving a lasting biogeochemical
imprint on surrounding waters (Arrigo et al., 1999; 2000; DiTullio et al., 2000; Dunbar et al.,
1998). Recent *in vitro* iron addition experiments provide evidence that iron nutrition influences *P.*
*antarctica* growth in this region, with increasing *P. antarctica* biomass after iron addition
(Bertrand et al., 2007; Feng et al., 2010). Moreover, laboratory experiments with *P. antarctica*
have observed a high cellular iron requirement and variable use of strong organic iron complexes
(Sedwick et al., 2007; Strzepek et al., 2011; Luxem et al., 2017).

The multiphasic lifecycle of *P. antarctica* in the Ross Sea gives it a spectrum of nutrient

drawdown phenotypes and trophic interactions, dependent on the presence of flagellated versus
colonial cells (Smith et al., 2003). Given its prominence during early spring sea ice retreat, it has
been hypothesized that the triggers of colony formation for *Phaeocystis* cells are also the triggers
of the spring phytoplankton bloom. Yet experimental and molecular analyses of potential
environmental triggers and how they manifest in changes in cellular morphology have remained
elusive. Little is known about the mechanisms responsible for colony formation in *P. antarctica*,
nor how these mechanisms respond to an environmental stimulus such as iron, both of which
appear to be integral to the ecology and biogeochemistry of *P. antarctica.*
**2. Materials and methods**
**2.1 Culture experiments**
Two strains of *Phaeocystis antarctica* (treated with Provasoli's antibiotics), CCMP 1871 and
CCMP 1374 (Provasoli-Guillard National Center for Culture of Marine Phytoplankton), and a
Ross Sea centric diatom isolate *Chaetoceros* sp. RS-19 (collected by M. Dennett at 76.5°S, 177.1°
W in December 1997 and isolated by D. Moran) were grown in F/2 media with a trace metal stock
(minus $FeCl_3$) according to Sunda and Huntsman (Sunda and Huntsman, 2003; 1995), using a
modified 10 μM EDTA concentration, and an oligotrophic seawater base. Strains were chosen
because they were culturable representatives from two distinct regions in the Southern Ocean.
Semi-continuous batch cultures were grown at 4 °C under 200 μmol photons $m^{-2}$ $s^{-1}$
continuous light. Each strain was acclimated to the six iron growth condition concentrations for at
least three transfers prior to proteome and growth rate experiments (> 9 generations per transfer
for > 27 generations). The concentration of dissolved inorganic iron within each treatment was 2
pM, 41 pM, 120 pM, 740 pM, 1200 pM, and 3900 pM Fe' as set by the metal buffer EDTA (where
$Fe'/Fe_{Total} = 0.039$) (Sunda and Huntsman, 2003). During the experiment, cultures were maintained
in 250 mL polycarbonate bottles; and, subsamples were collected every 1-2 days in 5 mL 13x100
mm borosilicate tubes to measure relative fluorescence units (RFUs) and cell counts in the
treatments. Mid-to-late exponential phase cultures were harvested for transcriptome and proteome
analysis and cell size was measured for both strains; cell pellets were stored at -80 °C (see
Supplementary Information for additional methods). Cell counts were conducted using a Palmer-
Maloney counting chamber and a Zeiss Axio Plan microscope on 400x magnification; cell
numbers were used to determine the final growth rate of each strain/treatment. During mid-to-late
exponential phase (time-of-harvest), cell size was determined for both strains (n=20 cells were
counted for each strain), calculated using the Zeiss 4.8.2 software and a calibrated scale bar. The
number of cells in colonies (versus as single cells) was determined for strain 1871 only. Briefly,
counts (number of cells associated with colonies versus unassociated) were averaged from 10
fields of view at five distinct time points (50 fields of view total).

**2.2 Protein extraction, digestion, and mass spectrometry analyses**
Proteins from cell pellets (one pellet per treatment, two strains and six iron treatments for a total
of 12 proteomes) was extracted using the detergent B-PER (Thermo Scientific), quantified,
purified by immobilization within an acrylamide tube gel, trypsin digested, alkylated and reduced,
and analyzed by liquid chromatography-mass spectrometry (LC-MS) using a Michrom Advance
HPLC with a reverse phase C18 column (0.3 x 10 mm ID, 3 μm particle size, 200 Å pore size,
SGE Protocol C18G; flow rate of 1 μL min$^{-1}$, nonlinear 210 min gradient from 5% to 95% buffer
B, where A was 0.1% formic acid in water and B was 0.1% formic acid in acetonitrile, all solvents
were Fisher Optima grade) coupled to a Thermo Scientific Q-Exactive Orbitrap mass spectrometer
with a Michrom Advance CaptiveSpray source. The mass spectrometer was set to perform MS/MS
on the top 15 ions using data-dependent settings (dynamic exclusion 30 s, excluding unassigned
and singly charged ions), and ions were monitored over a range of 380-2000 m/z (see
Supplementary Information for detailed protocol). Peptide-to-spectrum matching was conducted
using the SEQUEST algorithm within Proteome Discoverer 1.4 (Thermo Scientific) using the
translated transcriptomes for *P. antarctica* strain 1871 and strain 1374 (Fig 2., see below).
Normalized spectral counts were generated from Scaffold 4.0 (Proteome Software Inc.), with a
protein false discovery rate (FDR) of 1.0%, a minimum peptide score of 2, and a peptide
probability threshold of 95%. Spectral counts refer to the number of peptide-to-spectrum matches
that are attributed to each predicted protein from the transcriptome analysis, and the Scaffold
normalization scheme involves a small correction normalizing the total number of spectra counts
across all samples to correct for run-to-run variability and improve comparisons between
treatments. The R package "FactoMineR" (Lê et al., 2008) was used for the PCA analysis; for
heatmaps, the package "gplots" was used (Warnes et al., 2009). Proteomic samples taken from
each laboratory condition were not pooled downstream as part of the analyses; replicates shown
for each treatment are technical replicates.

**2.3 RNA extraction, Illumina sequencing, and annotation**
For *P. antarctica* cultures total RNA was isolated from cell pellets (one pellet per treatment, two
strains and three iron concentrations for a total of six transcriptomes) following the TRIzol Reagent
(Life Technologies, manufacturer's protocol). RNeasy Mini kit (Qiagen) was used for RNA
cleanup, and DNase I (Qiagen) treatment was applied to remove genomic DNA. Libraries, from
polyA enrichment mRNA, were constructed using a TruSeq RNA Sample Preparation Kit V2
(IlluminaTM), following the manufacturer's TruSeq RNA Sample Preparation Guide. Sequencing
was performed using the Illumina HiSeq platform. Downstream, reads were trimmed for quality
and filtered. CLC Assembly Cell (CLCbio) was used to assemble contigs, open reading frames
(ORFs) were predicted from the assembled contigs using FragGeneScan (Rho et al., 2010), and
additional rRNA sequences were removed.  The remaining ORFs were annotated de novo via
KEGG, KO, KOG, Pfam, and TigrFam assignments. Taxonomic classification was assigned to
each ORF and the Lineage Probability Index (LPI, as calculated in (Podell and Gaasterland, 2007).
ORFs classified as Haptophytes were retained for downstream analyses. Analysis of sequence
counts ("ASC") was used to assign normalized fold change and determine which ORFs were
significantly differentially expressed in pairwise comparisons between treatments. The ASC
approach offers a robust analysis of differential gene expression data for non-replicated samples
(Wu et al., 2010).
For metatranscriptomes, RNA was extracted from frozen cell pellets using the TRIzol
reagent manufacturer's protocol (Thermo Fisher Scientific) (see Supplementary Information for
additional details on metatranscriptome processing).

**2.4 Ross Sea *Phaeocystis* bloom: sample collection and protein extraction and analysis**
The meta 'omics samples were collected in the Ross Sea (170.76° E, 76.82° S) during the
CORSACS expedition (Controls on Ross Sea Algal Community Structure) on December 30, 2005
(near pigment station 137; http://www.bco-dmo.org/dataset-deployment/453377) (Saito et al.,
2010; Sedwick et al., 2011). Surface water was concentrated via a plankton net tow (20 μm mesh),
gently decanted of extra seawater, then split into multiple replicate cryovials and frozen in
RNAlater at -80 °C for metatranscriptome and metaproteome analysis. The pore size of the net
tow would have preferentially captured the colony form of *Phaeocystis*, although filtration with
small pore size membrane filters was particularly challenging during this time period due to the
abundance of *Phaeocystis* colonies and the clogging effect of their mucilage. Moreover, the
physical process of deploying the net tow appears to have entrained some smaller cells including
the *Phaeocystis* flagellate cells by adsorption to partially broken colonies and associated mucilage
as observed in the metaproteome results. Two of these replicate bloom samples were frozen for
proteome analysis. A third replicate sample from this field site was extracted for
metatranscriptome analysis as described above.
Proteins were extracted, digested, and purified following the lab methods above, and then
identified first on a Thermo Q-Exactive Orbitrap mass spectrometer using a Michrom Advance
CaptiveSpray source, then samples were subsequently re-run on a two-dimensional
chromatographic nanoflow system for increased metaproteomic depth on a Thermo Fusion
Orbitrap mass spectrometer (see supplemental materials for further details). Proteins were then
identified within the mass spectra using three databases (Fig. 2): the translated transcriptome
database for both *Phaeocystis* strains (Database #1), a Ross Sea metatranscriptome generated in
parallel from this metaproteome sample (Database #2; this transcriptome is a combination of
eukaryotic and prokaryotic communities derived from total RNA and poly(A) enriched RNA
sequencing), and a compilation of five bacterial metagenomes from the Amundsen Sea polynya
(Database #3) (Delmont et al., 2014), using SEQUEST within Proteome Discoverer 1.4 (Thermo
Scientific) (Eng et al., 1994) and collated with normalized spectral counts in Scaffold 4.0
(Proteome Software Inc.) (see Supplementary Information for additional details).

**2.5 Data availability**
*Phaeocystis antarctica* RNA sequence data reported in this paper have been deposited in the NCBI
sequence read archive under BioProject accession no. PRJNA339150, BioSample accession nos.
SAMN05580299 – SAMN05580303. Ross Sea metatranscriptomes have been deposited under
BioProject accession no. PRJNA339151, BioSample accession nos. SAMN05580312 –
SAMN05580313. Proteomic data from the lab and field components was submitted to the Pride
database (Project Name: *Phaeocystis antarctica* CCMP 1871 and CCMP 1374, Ross Sea
*Phaeocystis* bloom, LC-MSMS; Project accession: PXD005341; Project DOI:
10.6019/PXD005341).

**3. Results and discussion**
**3.1 Physiological response to iron availability: Growth limitation and colony formation**
The two strains of *P. antarctica* (1374 and 1871 hereon) were acclimated to six iron concentrations
to capture the metabolic response under different iron regimes (Fig. 3*a* and *b*). A biphasic response
in *P. antarctica* strain 1871 was observed; cultures exhibited a clear single-cell versus colony
response to low and high iron, respectively, that were observed by microscopy and were readily
apparent by naked eye due to the millimeter size of the colonies. The three low iron treatments (2
pM, 41 pM, and 120 pM Fe') cultures contained only single, flagellated cells, whereas the three
higher iron treatments (740 pM, 1200 pM, and 3900 pM Fe') had a majority of colonial cells,
based on detailed microscopy counts shown in Fig. 3*c*. This influence of iron on colony abundance
was observed in an additional experiment, where colonial cells were again absent at the lowest
three iron concentrations and were present at the three higher concentrations (Fig. S10). The
presence of both colony and flagellate cells is expected in actively growing populations since
reproduction can involve returning to the flagellate life cycle stage (Rousseau et al., 1994). Single
cells and colonies were not counted in experiments with strain 1374, as these experiments were
conducted prior to those of 1871 and the iron-induced colony formation observations therein.
However, strain 1374 was observed to become "clumpy" at high iron. This clumping observation
may reflect the loss of a specific factor needed for the colony completion lost during long-term
maintenance in culture. This interpretation is consistent with the overall similar structural protein
expression patterns observed in both strains described below. Strzepek et al. also observed co-
varying of iron concentration and colony formation in some strains of *P. antarctica* (Strzepek et
al., 2011).

The two strains of *P. antarctica* were able to maintain growth rates for all but the lowest

of iron concentrations used here, similar to prior studies of *P. antarctica* strain AA1 that observed
no effect of scarce iron on growth rates (Strzepek et al., 2011). Parallel experiments with polar
diatoms such as *Chaetoceros* (Fig. 3*d*) observed growth limitation at moderate iron abundances
using an identical media composition, indicating 1) that *P. antarctica* has an impressive capability
for tolerating low iron compared to *Chaetoceros* and other diatoms (e.g. a Ross Sea *Pseudo-*
*nitzschia* sp. isolate, data not shown), and 2) demonstrating an absence of iron contamination in
these experiments. Growth rates for 1871 were significantly different between the 2 pM Fe'
treatment and all other treatments (student's t-test with Bonferroni correction, $p < 0.05$; Fig. 3*a*);
there were no significant differences among growth rates for strain 1374 (Fig. 3*b*). Cell size
(including both flagellate and colonial cells) decreased with lower iron concentration, a trend that
was statistically significant (ANOVA with TukeyHSD test, $p < 0.05$) for both strains when cell
sizes from each high iron treatment (740 pM, 1200 pM, and 3900 pM Fe') were compared to cell
sizes from each low iron treatment (2 pM, 41 pM, and 120 pM Fe') (Fig. 3*e* and *f*).

**3.2 Molecular response to low and high iron concentrations**

Global proteomics enabled by peptide-to-spectra matching to transcriptome analyses,

revealed a clear statistically significant molecular transition across the iron gradient for each strain
(Fig. 4). The global proteome consisted of 536 proteins identified in strain 1871 and 1085 proteins
identified in strain 1374 (Table 1; Supplementary Data 1), after summing unique proteins across
the six iron treatments. There were 55 proteins identified in strain 1871 and 64 proteins in strain
1374 (Fig. 4) that drove the statistical separation of proteomes across iron treatments using
principle component analysis (PCA, Axis 1 PCA correlation coefficient ≥ 0.9 or ≤ -0.9). Axis one
accounted for 49% variance for 1871 and 36% variance for 1374. Moreover, using a Fisher Test
(P-value ≤ 0.05), 327 proteins (strain 1871) and 436 proteins (strain 1374) of those proteins
detected were identified as significantly different in relative protein abundance between
representative "low" (41 pM Fe') and "high" (3900 pM Fe') iron treatments. This significant
change in the proteome composition paralleled observations of a shift from flagellate to colonial
cells. Iron-starvation responses and iron metabolism were detected within the high and low iron
PCA protein subsets, including iron-starvation induced proteins (ISIPs), flavodoxin, and
plastocyanin, demonstrating a multi-faceted cellular response to iron scarcity (Fig. 5).
Surprisingly, there was also a highly pronounced signal in the proteome that appeared to reflect
the structural changes occurring in *P. antarctica*. These structural proteins included multiple
proteins with protein family (PFam) domains suggestive of extracellular function, adhesion, and/or
ligand binding, including putative glycoprotein domains (for example, spondin) that were present
in the high iron PCA subset in both strains (Fig. 5); the appearance of these proteins also
corresponded to the occurrence of colonies in strain 1871 (Fig. 1). Similarly, a distinct suite of
proteins was more abundant in the low iron PCA subset (Fig. 5), including proteins relating to cell
signaling (for example, calmodulin/EF-hand, PHD zinc ring finger). A number of proteins with
unknown function were also detected in the PCA subsets: 71% unknown for strain 1871 and 42%
unknown for strain 1374 of a total of 311 proteins annotated as hypothetical proteins
(Supplementary Dataset 1). Outside of the PCA analyses, additional iron and adhesion-related
proteins were identified that demonstrated a similar expression profile to the PCA subset
(Supplementary Fig. 1).

Identification and characterization of proteins and transcripts induced by iron scarcity are

valuable in improving an understanding of the adaptive biochemical function of these complex
phytoplankton as well as for their potential utility for development as environmental stress
biomarkers (Roche et al., 1996; Saito et al., 2014). The enzymes flavodoxin and plastocyanin,
which require no metal and copper, respectively and that functionally replace iron metalloenzymes
counterparts ferredoxin and cytochrome c6, had isoforms that increased in concentrations at the
lower iron treatments consistent with cellular iron sparing strategies (Fig. 6, Supplementary Fig.
2) (Peers and Price, 2006; Whitney et al., 2011; Zurbriggen et al., 2008). In strain 1374 however,
there was an increase in both of these iron-sparing systems at the highest iron concentration (Fig.
6d and 6f, Supplementary Table 1). While during both experiments, cells were growing
exponentially at the time of harvest, those of strain 1374 were as much as 7.6 fold denser in cell
number than those of strain 1874 (based on cell counts from treatments specifically used for
transcriptome analyses), and as a result the denser 1374 strain might have also experienced iron
stress even at this highest iron concentration as the high biomass depleted iron within the medium.
Of these two iron sparing enzymes, plastocyanin appeared to show a clearer increase in abundance
at lower environmental iron concentrations (Fig 6c and 6f). In contrast, some flavodoxin isoforms
could be interpreted as being constitutive, two of the three isoforms were still present in reasonable
spectral counts at higher iron concentrations (Figs. 6a and 6d). Prior measurements during a Ross
Sea colonial *P. antarctica* spring bloom in 1998 were consistent with this interpretation, with
ferredoxin concentrations were below detection and flavodoxin present (Maucher and DiTullio,
2003). A constitutive flavodoxin could help explain *P. antarctica's* ability to tolerate all but the
lowest iron treatment observed in the physiological experiments (Fig 3a and 3b), and implies that
the careful selection of isoforms, or better, the inclusion of all isoforms of a protein biomarker of
interest may be valuable in interpreting complex field results.
There were also numerous isoforms of the iron-starvation induced proteins (ISIP) group
identified within the proteome of each *P. antarctica* strain: specifically 9 ISIP2A's and 3 ISIP3's
in strain 1871 and 3 ISIP2A's and 4 ISIP3's in strain 1374 (Supplementary Fig. 1; Supplementary
Table 1). These ISIPs were identified based on their transcriptome response to iron stress in
diatoms and most recently have been implicated in a diatom cell surface iron concentrating
mechanism (Allen et al., 2008; Morrissey et al., 2015). Interestingly in this *P. antarctica*
experiment, these ISIPs exhibited both "high" or "low" iron responses, where specific isoforms
were more abundant only under one of those respective conditions (Fig. 6). Given the
metamorphosis of *P. antarctica* between flagellate and colonial cell types observed by microscopy
and the proteome across the gradient in iron concentrations, we hypothesize that this diversity of
iron stress responses in the ISIP proteins may reflect the complexity associated with *P. antarctica*'s
life cycle. As the abundant winter iron and sloughed basal sea ice reserves are depleted, newly
formed colonial cells will inevitably find themselves in the iron-depleted environments that have
been characterized in the Ross Sea almost immediately upon bloom formation due to iron's small
dissolved inventory (Bertrand et al., 2015; Sedwick et al., 2011). As a result, *P. antarctica* may
have distinct iron stress protein isoforms associated specifically with the colonial cell type (such
as the high iron/colonial ISIP proteins, Figs. 5 and 6) in order to acquire scarce iron during blooms,
in addition to a distinct suite of iron stress proteins produced within the flagellate cells (low
iron/flagellate ISIP proteins, also Figs. 5 and 6). Given the rapid depletion of iron during Ross Sea
blooms, it is also conceivable that these iron acquisition proteins are constitutively expressed
within the colony morphotype, rather than being connected to an iron-sensing and regulatory
response system. Future short-term iron perturbation studies that would complement the steady-
state experiments presented here could further investigate this hypothesis. The multiplicity of ISIP
proteins produced within each strain also is consistent with the observation that both *P. antarctica*
strains maintained high growth rates even at the lower 41 and 120 pM Fe' concentrations,
compared to the diatom *Chaetoceros sp.* whose growth rate is less than 50% of maximal growth
in similar media (Fig. 3).

**3.3 Correspondence between RNA and protein biomolecules**

Many of the RNA transcripts of iron-related genes trended with their corresponding

proteins: 60% of the iron-related gene transcripts reflected the proteomic response in strain 1871,
whereas there was a 30% correspondence between iron-related transcripts and proteins in strain
1374 (Supplementary Fig. 1). In total, 47% of expressed proteins in strain 1871 and 26% of
proteins in strain 1374 shared expression patterns with associated transcripts (Fig. 7), consistent
with recent studies of proteome-transcriptome comparisons that showed limited coordination
between inventories of each type of biomolecule (Dyhrman, 2012). As mentioned above, while
both experiments were in exponential growth at the time of harvest, strain 1374 was 7.6 fold denser
in cell number than those of strain 1874 at that time. Hence, this decrease in transcript-proteome
coherence in strain 1374 may be related to harvesting in late-log growth phase, and reflects the
challenge of trying to conduct comparisons of these biomolecules that function on different cellular
timescales.

Examination of the transcriptome revealed a significant increase in transcripts for tonB-

like transporters, which can be associated with cross-membrane nutrient transport (e.g. for iron
siderophores complexes or vitamin $B_{12}$ (Bertrand et al., 2007; 2013; Morris et al., 2010) under high
iron for strain 1871; and, significantly greater transcript abundances for a putative flavodoxin for
strain 1374 under low iron consistent with its substitution for ferredoxin due to iron scarcity (Roche
et al., 1996).

**3.4 Observation of an iron-induced switch from single cells to colonies**

The strong connection of iron availability to putative structural components of *P.*

*antarctica* observed here served as an ideal opportunity to examine the genes and proteins involved
in morphological and life cycle transitions and colony construction in this phytoplankter that can
otherwise be experimentally difficult to trigger in isolation. *Phaeocystis* colonies have captured
the interest of scientists for more than a century (Hamm et al., 1999), yet next to nothing is known
about the molecular basis of their construction. Colonies have been considered a collection of
loosely connected cells embedded within a gel matrix, and hence described as "balls of jelly" or
"bags of water" (Hamm et al., 1999; Lagerheim, 1896; Verity et al., 2007). Results here suggest
significant transformations in the cellular proteome that corresponded to solitary and colonial
morphological stages, for example, involving structural proteins and proteins known to be post-
translationally modified such as glycoproteins or those containing glycoprotein-binding motifs. To
our knowledge, such an extensive proteome remodeling has yet to be observed for another colonial
organism, or with the influence of any other environmental stimuli in the genus *Phaeocystis*. As a
result the details of this response, while fascinating, are challenging to interpret due to their
novelty.

A putative spondin protein exhibited one of the largest responses between low and high

iron in both strains with a greater than 20-fold increase in relative protein abundance and
normalized 11-fold change in transcript abundance in strain 1871, and a greater than 9-fold
increase in relative protein abundance and 3-fold change in transcript abundance in strain 1374
(Fig. 5*a* and Supplementary Data 1). Spondin proteins are known to be glycosylated, and to be a
component of the extracellular matrix (ECM) environment, which may enable multicellularity in
metazoans through cell adhesion, and have been found to help coordinate nerve cell development
through adhesion and repulsion (Michel et al., 2010; Tzarfati-Majar et al., 2011). Despite this large
variation in protein abundance, the function of spondins in eukaryotic phytoplankton, including
*Phaeocystis* remains largely unknown. Given their responsiveness to iron availability and
associated enrichment in colony rich cultures, these proteins could potentially contribute to ECM-
related adhesion of cells, to each other or the colony skin, or even perhaps to the mucilage interior.
Additional glycoproteins that exhibited a strong iron response in both strains include those
containing von Willebrand factor domains (for example, protein families PF13519, PF00092), and
fibrillin and lectin (Fig. 5 and Supplementary Fig. 1). In biomineralizing organisms, such as corals,
glycoproteins with von Willebrand domains are hypothesized to play a role in the formation of the
extracellular organic matrix through adhesion (Drake et al., 2013; Hayward et al., 2011) laying the
scaffolding for calcification. Orthologs of the von Willebrand proteins that contain these domains
have also been characterized in humans and have protein-binding capabilities, which are important
for coagulation (Ewenstein, 1997). These dynamic von Willebrand proteins appear to contribute
to the cell aggregation and colony formation of *P. antarctica* colonies.
The suite of structural and modified proteins described above demonstrates a means
through which *P. antarctica*'s colonial morphotype could be constructed, and this dataset provides
rare molecular evidence for the proteome reconstruction needed to switch between single
organisms to a multicellular colony. The evolution of multicellularity in Eukaryotes is an area of
significant interest that has mostly focused on model organisms with colonial forms such as
Choanoflagellates and *Volvox* (Abedin and King, 2010). Genomic studies of the former identified
the presence of protein families involved in cell interactions within metazoans, including C-type
lectins, cadherins, and fibrinogen (King et al., 2003). In other lineages of microalgae that form
colonial structures, such as *Volvox carteri*, there is supporting evidence for glycoproteins cross-
linking within the extracellular matrix of colonies (Hallmann, 2003), as well as serving other
important functional roles in cell-cell attachment during colony formation (for example, colony
formation in the cyanobacteria *Microcystis aerginosa*) and as an integral component of cell walls
(for example, the diatoms *Navicula pelliculosa* and *Craspedostauros australis*) (Chiovitti et al.,
2003; Kröger et al., 1994; Zilliges et al., 2008). In this study, environmental isolates of *P.*
*antarctica* displayed consistent trends in similar protein families (for example, lectins, fibrillins,
and glycoproteins), and they were produced at higher levels under elevated iron conditions when
strain 1871 was primarily in colonial form. Given *P. antarctica*'s environmental importance and
an ability to control the transition between flagellates and colony cell types through iron
availability, *P. antarctica* may serve as a useful model for studying multicellularity in nature and
in the context of environmental change.

In contrast to these putative colonial structural proteins, there were canonical cytoskeletal

proteins such as actin and tubulin observed in *P. antarctica* cultures grown under low iron
conditions (Supplementary Fig. 1). These proteins were likely associated with the flagella and the
haptonema, a shorter organelle containing nine microtubules that is characteristic of Haptophytes
(Zingone et al., 1999), found in the solitary *Phaeocystis* cell, and similar to other eukaryotic
flagellar systems such as *Chlamydomonas* (Watanabe et al., 2004). Additionally, a suite of proteins
with calcium-binding domains (EF-hand protein families) was identified in greater relative
abundance under low iron growth conditions in both strains (Fig. 5; Supplementary Fig. 1 and
Supplementary Data 1). In diatoms, calcium-signaling mechanisms have been directly linked with
how cells respond to bioavailable iron, as well as stress responses (Allen et al., 2008; Vardi, 2008).
Calcium (and magnesium) ions also play an integral role in the ability for extracellular mucus to
gel (van Boekel, 1992). The greater abundance of putative calcium-binding proteins under low
iron conditions suggests an important role for intracellular calcium, either in its involvement in
flagellate motility and/or having a role in inhibiting the cells' abilities to form colonies while under
iron limitation. This use of calcium signaling is notable given that calcium is a major constituent
of seawater (0.01 mol $L^{-1}$), implying a need for efflux and exclusion of calcium from the
cytoplasm.

**3.5 *Phaeocystis antarctica* strain-specific responses**
*Phaeocystis antarctica* is believed to have speciated from warm-water ancestors, and populations
within the Antarctic are mixed via the rapid Antarctic Circumpolar Current (ACC, 1-2 years)
circulation with the Ross Sea and Weddell Sea, which entrains strains nearly simultaneously
(Lange et al., 2002). Moreover, high genetic diversity has been observed across a large number of
*P. antarctica* isolates and even within isolates co-isolated from a bloom (Gäbler-Schwarz et al.,
2015). Given the differences in geographic location of the isolates used in this study, there may be
some differences regarding adaptation and ecological role between them. In the Ross Sea, *P.*
*antarctica* dominates, and cells exhibit seasonal variability between flagellated states (early
Spring, late summer) and colonial stage (late Spring/early summer) (Smith et al., 2003). In
contrast, in the Western Antarctic Peninsula, near the Weddell Sea where strain 1871 was isolated
from (Palmer station), *P. antarctica* is outnumbered by diatoms and cryptomonads in terms of
algal biomass, and colonies are generally rare (Ducklow et al., 2007). While global proteomic and
transcriptomic analyses revealed differences among strains (Supplementary Data 1), both strains
had responses that overwhelmingly supported a concerted effort towards structural changes under
high iron versus low iron, consistent with the minor phylogenetic differences previously reported
for *P. antarctica* isolates due to rapid ACC circulation (Lange et al., 2002).

**3.6 Examination of a *Phaeocystis* bloom metaproteome from the Ross Sea**

The detailed laboratory studies above can be compared to a first metaproteomic analysis

of a Ross Sea *Phaeocystis antarctica* bloom to provide an examination of the *in situ* ecology and
biogeochemical and their underlying biochemical signatures. Due to the newness of
metaproteomic eukaryotic phytoplankton research, some methodological detail has been
incorporated into this section. For field analysis a net tow sample was collected north of Ross
Island (Fig. 8) on December 30[th] 2005, in which *Phaeocystis* colonies were visually dominant.
Temporal changes in the bloom composition have been described for this summer expedition and
an austral spring expedition later that year (NBP06-01 and NBP06-08, respectively), and a shift
was observed from a *P. antarctica* dominated ecosystem to a mixture of *P. antarctica* and diatoms
(Smith et al., 2013). Surface pigment distributions showed the sampling region to be within a
particularly intense bloom dominated by *Phaeocystis* as observed by abundant 19'-
hexanoyloxyfucoxanthin pigment (Fig. 8), reaching concentrations of 1096 ng L$^{-1}$ and total
chlorophyll *a* concentrations of 1860 ng L$^{-1}$ on the sampling day. CHEMTAX analysis of these
HPLC pigments found that *P. antarctica* populations accounted for approximately 88% of surface
water total chlorophyll at this time. Fucoxanthin pigment, characteristic of diatoms, was lower
here (136 ng L$^{-1}$) compared to samples from the western Ross Sea (Fig. 8), consistent with prior
Ross Sea observations. Repeated sampling near the sampling region (~77.5$^{o}$S) two weeks after
taking the metaproteome sample found lower overall chlorophyll *a* levels (Smith et al., 2013),
consistent with bloom decay. Iron measured very near this location (76.82º S, 170.76º E also on
December 30, 2005), found a surface dissolved iron concentration of 170pM (6m depth) and an
acid-labile particulate iron concentration of 1590 pM (Sedwick et al., 2011), consistent with iron
depletion in seawater following drawdown of the accumulated winter iron supply and
incorporation of iron into biological particulate material (Noble et al., 2013; Sedwick et al., 2000).

The metaproteome analyses of the Ross Sea sample were conducted by bottom-up mass

spectrometry analysis of tryptic peptides using initially a 1-dimension and subsequently a deeper
2-dimension chromatographic methodology (1D and 2D hereon), followed by peptide-to-spectrum
matching of putative peptide masses and their fragment ions to predicted peptides from translated
DNA sequences. While this approach is common for model organisms and has been successfully
applied to primarily prokaryotic components of natural communities (Morris et al., 2010; Ram et
al., 2005; Sowell et al., 2008; Williams et al., 2012), there continue to be challenges in
metaproteomics analyses of diverse communities particularly when including an extensive
eukaryotic component such as is present in the Ross Sea phytoplankton bloom. VerBerkmoes et
al. (2005) demonstrated the feasibility of using mass spectrometry metaproteomic analysis for the
detection of eukaryotic proteins in a complex sample matrix. To address these issues, we utilized
three sequences databases for peptide-to-spectrum matching (see Methods and Supplementary
Information Table S2). Analysis of both unique (tryptic) peptides and identified proteins are
provided here, where unique peptides are particularly valuable in metaproteome interpretation as
a basal unit of protein diversity that can be definitively compared across the three sequence
databases (Saito et al., 2015).
The combined *P. antarctica* strain transcriptome database (Database #1) generated the
largest number of protein and unique peptide identifications 1545 and 3816 in 2D, (912 and 2103
in 1D), respectively (Table 2, Fig. 9*a*). This strong relative performance of the strain database was
surprising, and likely reflects the depth of the *P. antarctica* isolate transcriptomes and resultant
translation into greater metaproteomic depth. Approximately sixty percent of field identifications
mapped to strain 1374 (57%); a broad synthesis of all proteomes based on KOG annotations also
indicated that the metaproteomes appeared most similar to the Ross Sea strain 1374
(Supplementary Fig. 3). The Ross Sea metatranscriptome database (Database #2) resulted in 1475
proteins and 3210 unique peptides in 2D analyses (859 proteins and 1520 unique peptides in 1D)
distributed across a large number of taxa, with 324 of those proteins associated with *P. antarctica.*
The Antarctic bacterial metagenome database (Database #3) produced 102 proteins and 237 unique
peptides in 2D (98 proteins and 186 peptides in 1D) that mapped to bacteria likely associated with
the phytoplankton communities, given the use of a net that would not otherwise capture free-living
bacteria. The low number of bacterial protein and peptides identifications could reflect their small
abundance or limited metagenomic coverage. Due to the extensive diversity present, there was
overlap between the peptide identifications from each database for the 5885 total unique peptides
in2D (3193 in 1D) : 1222 (in 2D; 544 in 1D) *P. antarctica* peptides were shared between the
*Phaeocystis* strain and Ross Sea metatranscriptome databases, 158 (in 2D; 69 in 1D) bacterial
peptides were in common between the Ross Sea metatranscriptome and the bacterial metagenomic
databases, followed by very small numbers shared between bacterial metagenome and the
*Phaeocystis* strains database searches (8 peptides in both 1D and 2D), and all three databases (7
and 4 peptides in 1D and 2D, respectively), likely due to a small fraction of tryptic peptides shared
between diverse organisms (Saito et al., 2015).

This multi-database approach and the relatively low overlap illustrates the necessity of

employing diverse sequence databases that target distinct components of the biological
community, as well as the value in coupling metatranscriptomic and metagenomic sequence
databases to metaproteomic functional analysis to capture the extent of natural diversity. This is
evident in the taxon group analysis, where the metatranscriptome has a large representation of
Dinophyta and diatoms and only a small contribution from Haptophyta that include *Phaeocystis*,
likely due to the large genome sizes and transcription rates, particularly of dinoflagellates, and
perhaps due to interferences of *Phaeocystis* RNA extraction due to the copious mucilage present
(Fig. 9*b*). In contrast the metaproteome derived from the metatranscriptome database is dominated
by Haptophyta and Dinophyta, with minor contributions from other groups (Fig. 9*d*), reflecting
the dominant organismal composition seen in the pigment analyses (Fig. 8). Due to a coarse net
mesh size much larger than a typical bacterial cell, the bacterial community captured by these
metatranscriptome and metaproteome analyses most likely reflects the microbiome associated with
larger phytoplankton and protists, particularly within the abundant *P. antarctica* colonies.
Database #2 and #3 result in 211 and 102 bacterial protein identifications (in 2D; 148 and 100 in
1D), respectively, including representatives from *Oceanospirillacea*e, *Rhodobacteraceae*,
*Cryomorphaceae, Flavobacteria*, and *Gamma proteobacteria* (Fig. 9*c* and *d*). The lower number
of bacterial identifications could be due to low bacterial biomass in the net tow sample relative to
phytoplankton biomass and/or limited metagenomics coverage.

Together this Ross Sea bloom metaproteome-metatranscriptome analysis provides a

window into the complex interactions of this community with its chemical environment.
*Phaeocystis antarctica* proteins were abundant in the sample with over 450 (in 2D; 300 in 1D)
proteins identified, yet interestingly, we identified proteins associated with both high and low iron
treatments, including those corresponding to flagellate and colonial life stages identified in the
culture experiments (Fig. 10 and Supplementary Fig. 1). This presence of both life cycle stages of
*Phaeocystis* could be interpreted as evidence of an actively growing bloom, with growing
flagellate cells coalescing to form new colonies, as well as a standing stock of colonial cells. As
mentioned earlier, division and growth of *P. antarctica* colonies is believed to require transitioning
back through the flagellate life cycle stage (Rosseau et al., 1994), hence a mixed population of
flagellate and colonial stages would be expected of a growing population, consistent with our
laboratory observations (Fig. 3*c*).

The presence of well-known iron-sparing proteins such as plastocyanin (Fig. 10) was

consistent with the depleted dissolved iron concentration (170 pM) in nearby surface waters that
are closest to the 120 pM Fe' of the low iron treatments (Peers and Price, 2006; Sedwick et al.,
2011), as well as incubation experiments on the same expedition initiated three days prior that
demonstrated iron limitation of *P. antarctica* (and iron-$B_{12}$ colimitation of diatom) populations
(Bertrand et al., 2007).  Notably, the actual Fe' of the Ross Sea was likely considerably lower than
this due to the presence of strong organic iron complexes (Boye et al., 2001). Strzepek et al. found
evidence for growth of *P. antarctica* and some polar diatoms on strong organic iron complexes at
somewhat reduced growth rates in their culture experiments, implying a high-affinity iron
acquisition system such as a ferric reductase, although the molecular components of such a system
have yet to be identified in *P. antarctica* (Strzepek et al., 2011). As described above, it is likely
that both flagellate and colonial cell types have a need to manifest iron stress responses (e.g.
distinct ISIP proteins found in the flagellate and colonial dominated cultures, Figs. 5 and 6), and
that those distinct responses may be based on the extensive physical differences between life cycle
phenotypes. The low contribution of chain-forming diatoms to this metaproteome sample was
consistent with the higher sensitivity of some Ross Sea diatom strains to iron stress such as
*Chaetoceros* (Fig. 3*d*) and the low iron availability. Careful examination of targeted mass
spectrometry results (precursor and fragment ion analysis) for select iron proteins identified in
culture studies showed consistently high quality chromatograms within the field sample,
demonstrating a capability to measure these potential peptide biomarkers within complex
environmental samples in future field studies characterizing bloom and biogeochemical dynamics
(Fig. 11 and Supplementary Figs. 4-10).

The metaproteome analyses also captured relevant functional elements of the bacterial

microbiome associated with the eukaryotic community, based on the bacterial proteins identified
in both the bacterial databases and the Ross Sea metatranscriptome (Fig. 9*c* and 9*d*). For example,
the SAR92 clade of proteorhodopsin-containing heterotrophic bacteria was present (Stingl et al.,
2007), and expressed both the iron storage protein bacterioferritin and TonB receptors, the latter
of which are involved in siderophore and $B_{12}$ transport. In addition, the Fur iron regulon, iron-
requiring ribonucleotide reductase, as well as the vitamin related CobN cobalamin biosynthesis
protein, $B_{12}$-requiring methyl-malonyl CoA, and thiamine ABC transporter were observed from
several heterotrophic bacteria species including *Oceanospirillacea*e, *Rhodobacteraceae*, and
*Cryomorphaceae* (Supplementary Data 2) (Bertrand et al., 2015; Murray and Grzymski, 2007).
These results imply that heterotrophic bacteria known to be associated with the *Phaeocystis*
colonies, such as SAR92 and *Oceanospirillaceae*, were also likely responding to micronutrients
by concentrating and storing iron, and through biosynthesis of $B_{12}$. In doing so this bacterial
microbiome could have been harboring an "internal" source of the micronutrients, fostering a
mutualism with *Phaeocystis* colonies in exchange for a carbon source and consistent with the high
particulate iron measured during this station (Sedwick et al., 2011). Together this could create a
competitive advantage for *P. antarctica* relative to the iron and $B_{12}$-stressed diatoms for early
season bloom formation, as previously hypothesized and observed in the Ross Sea in enrichment
studies (Bertrand et al., 2007). Although diatoms were less prominent in the dataset, several diatom
proteins identified were indicative of the potential for iron stress (e.g., plastocyanin and ISIP3;
Supplementary Data 2); however, the diatom CBA1 cobalamin acquisition protein was not
identified in the metatranscriptome, and hence would not be detected in the metaproteome using
the current methods, but could be targeted in future studies from this dataset.

**4.  Conclusions**
*Phaeocystis antarctica* is a major contributor to Southern Ocean primary productivity, yet
arguably is one of the least well understood of key marine phytoplankton species. The multiple
life cycle stages of *P. antarctica* add to its ecological and biochemical complexity. Here we have
undertaken a detailed combined physiological and proteomic analysis enabled by transcriptomic
sequencing under varying conditions of iron nutrition, and compared these to an initial study of
the metaproteome of a Ross Sea *Phaeocystis* bloom. These results demonstrate that *P. antarctica*
has evolved to utilize elaborate capabilities to confront the widespread iron scarcity that occurs in
the Ross Sea and Southern Ocean, including iron metalloenzyme sparing systems and the
deployment of transport and other systems that appear to be unique to the flagellate and colonial
morphotypes. To our surprise, increasing iron abundance triggered colony formation in one strain
in this study, and visual and proteomic evidence implied the second strain was also attempting to
do so. Prior studies have invoked light irradiance and mixed layer depth as key factors in colony
production and the concurrent Ross Sea *P. antarctica* bloom initiation (Arrigo et al., 1999), and
hence there may be other factors that could have this effect as well. These results also provide
preliminary insight into the cellular restructuring processes that occurs upon cellular
metamorphosis between life cycle stages in *P. antarctica,* as well as identifying numerous dynamic
proteins of unknown function for future study. Finally, this study demonstrates the potential for
the application of coupled transcriptomic and proteomic biomarker methodologies in studying the
ecology of microbial interactions (including iron and $B_{12}$) and their influence on biogeochemistry
in complex polar ecosystems such as the Ross Sea. The improved molecular and biochemical
understanding of *P. antarctica* and its response to iron provided here are valuable in the design of
future experiments and targeted metaproteomic assays to examine natural populations and to
improve understanding of environmental factors that influence the annual bloom formation of an
important coastal ecosystem of the Southern Ocean.


**Acknowledgements**
Support for this study was provided by an Investigator grant to M. Saito from the Gordon and
Betty Moore Foundation (GBMF3782), and National Science Foundation grants NSF-PLR
0732665, OCE-1435056, OCE-1220484, the WHOI Coastal Ocean Institute, and a CINAR
Postdoctoral Scholar Fellowship provided to S. Bender through the Woods Hole Oceanographic
Institution. Support was provided to A. Allen through NSF awards ANT-0732822, ANT-1043671,
OCE-1136477, and Gordon and Betty Moore Foundation grant GBMF3828. Additional support
was provided to GRD through NSF award, OPP-0338097. We are indebted to Roberta Marinelli
for her leadership and vision. We would also like to thank Emily Lorch for her assistance with
culturing, Julie Rose for generously sharing a net tow field sample, and Andreas Krupke for
manuscript feedback.

**Author Contributions**

S.J.B. contributed to data analysis and writing; D.M.M. conducted the laboratory experiments and (meta)proteome extractions; M.R.M. conducted the mass spectrometry sample preparation and processing; H.Z. conducted RNA extractions; J.P.M. and J.B. contributed to transcriptome sequence analyses; G.R.D. contributed to field measurements and manuscript edits; A.E.A. contributed to the experimental design, data analysis, and writing; M.A.S. contributed to experimental design, data analysis, and writing.

**Financial Conflicts:** The authors have no financial conflicts involving the research presented in this manuscript.

650

**Table 1.** Comparison of the total number of proteins and spectra measured in the proteome for each strain/treatment along with the number of differentially expressed transcripts between select conditions for *P. antarctica* strain 1871 and strain 1374. Proteins were identified using a 1% FDR (false discovery rate) threshold, a peptide threshold of 95%, and a minimum of 2 unique peptides per protein. The total number of peptide-to-spectrum matches (PSMs) is given for the total of each strain in parentheses. A threshold of 3 spectral counts in at least one of the treatments was selected for inclusion in the comparative analysis.


| Strain | Treatment (Fe' pM) | Proteins Identified (PSMs) |
|--------|--------------------|----------------------------|
| 1871 | 2 | 204 |
| | 41 | 214 |
| | 120 | 234 |
| | 740 | 226 |
| | 1200 | 251 |
| | 3900 | 258 |
| | Total | 536 (28887) |
| 1374 | 2 | 581 |
| | 41 | 613 |
| | 120 | 600 |
| | 740 | 654 |
| | 1200 | 623 |
| | 3900 | 527 |
| | Total | 1085 (72087) |

**Table 2.** Comparison of the total number of proteins, peptides, and spectra measured in the
Ross Sea metaproteome net tow sample using three databases for peptide-to-spectrum
matching (see Table S2). Results from 2-dimension and 1-dimension (1D in parentheses)
analyses are shown.

| Peptide-to-spectrum - matching database | Total proteins | Total Unique Peptides | Total spectra matched | Decoy FDR[+] Percent (peptide level) |
|---|---|---|---|---|
| *1) Phaeocystis* strains transcriptomes* | 1545 (912) | 3816 (2103) | 14088 (8226) | 0.6 (0.17) |
| 2) Ross Sea metatranscriptome** | 1474 (859) | 3210 (1520) | 10154 (4725) | 0.1 (0.7) |
| 3) Antarctic bacterial metagenomes*** | 102 (92) | 237 (186) | 530 (440) | 3.6 (2.3) |


[+]FDR refers to false discovery rate of a reversed peptide database
* Metaproteome annotated using the laboratory-generated transcriptomes for strain 1871
and strain 1374 (database #1).
** Metaproteome annotated using the metatranscriptome generated from sample split of
original Ross Sea sample (database #2).
*** Bacterial metaproteome annotated using bacterial metagenomes from Delmont et al.,
2014 (database #3).

**Figure Legends**

**Figure 1.** Micrographs of (a) a single *Phaeocystis* in cell culture, and (b) *Phaeocystis* colonies in a Ross Sea bloom.

**Figure 2.** Experimental workflow used in this study. Culture and field samples (top), transcriptome analyses (2nd row), sequence database construction for proteomics (3rd row), and proteomic and metaproteomic analyses (bottom row).

**Figure 3.** The effect of iron concentration on colony formation and cell physiology in two strains of *P. antarctica* – 1871 and 1374. Growth rates collected from acclimated culture stocks prior to the start of the experiments (*a*, strain 1871; *b*, strain 1374), calculated using relative fluorescence units from three transfers of acclimated cultures (error bars indicate SD, n=3). Accompanying gray bars represent growth rates calculated based on cell counts made during the course of the proteome-harvest experiments (n=1). (*c*) The number of *P. antarctica* 1871 free-living cells (gray bars) compared to cells associated with colonies (black bars) showed a shift to a majority of colonial cells when Fe' $\geq$ 740 pM. (*d*) Growth rate of Ross Sea diatom isolate *Chaetoceros* sp. strain RS-19 in the same media compositions (n=1), demonstrated a higher sensitivity to iron scarcity and a lack of iron contamination in the media. Cell size for strain 1871 (*e*; black circles) and strain 1374 (*f*; white circles); error bars represent SD of n=20 cell measurements per treatment.

**Figure 4.** Principle Component Analysis (PCA) of the measured proteomes for each iron condition for strain 1871 and strain 1374 and corresponding line graphs highlighting the proteins

driving the PCA separation (PCA analyses: ≥ 0.9 or ≤ -0.9). (*a* and *d*) Iron treatments (pM Fe')
are highlighted by color (2, black; 41, red; 120, orange; 740, green; 1200, purple; 3900, blue) and
large ellipses indicate confidence ellipses calculated using the R package, FactorMineR. Each
small, solid circle represents a technical replicate per iron treatment (n=3); colored, open squares
represent the mean of the iron treatment (empirical variance divided by the number of
observations). Proteins with Eigen values ≥ 0.9 or ≤ -0.9 are plotted in graphs b and c for strain
1871 and *e* and *f* for strain 1374 to highlight the subset of proteins driving the variance in
Dimension 1. Individual protein spectral counts normalized to total spectral counts for all
treatments for a given protein, written as "normalized relative protein abundance" are plotted on
the y-axis. The six iron treatments (pM Fe') are plotted from low to high (left to right) on the x-
axis.

**Figure 5.** Heatmaps highlighting the relative protein abundance for the six treatments for *P.*
*antarctica* strain 1871 (*a*) and strain 1374 (*b*). The darker green color indicates a greater relative
abundance compared to the purple treatments. The "shared abundance patterns" column features
a check-mark when a shared response to changes in iron availability between the relative protein
abundance and the transcript abundance was observed (for example, both transcripts and proteins
have a higher abundance under high iron compared to low iron growth [or] both transcripts and
proteins have a higher abundance under low iron compared to high iron growth). The "field
presence" column indicates whether or not that protein was detected in the field metaproteome
(annotated using Database #1). Protein annotations are based on KEGG, KOG, and PFam
descriptions. Annotations in red are associated with iron metabolism and those in blue, cell
adhesion/structure.

**Figure 6.** Examination of iron stress response proteins in *P. antarctica* strain 1871 (top) and 1374 (bottom). Relative protein abundance is shown as normalized spectral counts, where spectral counts have been normalized across experiment treatments for each strain, but not to the maximum of each protein as used in prior figures to allow comparison of abundance for similar isoforms. Error bars indicate the standard deviation of technical triplicate analyses.

**Figure 7.** Scatterplots of relative transcript abundance (y-axis) and relative protein abundance (x-axis) for *P. antarctica* strain 1871 (a) and strain 1374 (b) for a high iron treatment (3900 pM Fe') relative to a low iron treatment (41 pM Fe'). Gray circles represent instances where transcript abundance was not significantly different between conditions ($P \geq 0.99$). Quadrants where relative protein and transcript abundances agree (upper right, lower left) and disagree (upper left, lower right) are noted, as are select genes exhibiting the greatest relative protein abundance and/or transcript abundance under a given treatment.

**Figure 8.** Location of the metaproteome sample and pigment data from a Ross Sea *Phaeocystis* bloom net tow sample. (*a*) Station map of NBP06-01 (December 27, 2005 to January 23, 2006) and the metaproteome sample was taken on December 30[th] by net tow location (red circle). (*b*) 19'-hexanoyloxyfucoxanthin ("19'-Hex") pigment is associated with *Phaeocystis*, while (c) peridinin and (d) fucoxanthin pigments are typically associated with dinoflagellates and diatoms, respectively (although dinoflagellates living heterotrophically can be lacking in pigment). Comparisons of the spring and summer expeditions (NBP06-08 and NBP06-01, respectively),

observed a shift from being dominated by *P. antarctica* to being a mixture of *P. antarctica* and
diatoms. See Smith et al., 2013 for further details (Smith et al., 2013).

**Figure 9.** (a) Venn diagram of the attribution of the5885 total unique peptides identified in the
metaproteome sample to three DNA/RNA sequence databases (Supplementary Table 2). (*b*)
Taxon group composition of genes identified by metatranscriptome analyses (combining Total
RNA and PolyA RNA fractions). (c) Taxon group composition of proteins identified by the
bacterial metagenomic database (Database #3). (d) Taxon group composition of proteins
identified by metatranscriptome database (Database #2).

**Figure 10.** Putative biomarkers identified in the *Phaeocystis* metaproteome annotated using the
field metatranscriptome (error bars represent SD of replicate samples; n=2; 1D dataset used).
Green bars indicate putative "low iron" biomarkers; red bars indicate putative "high iron"
biomarkers, and correspond to the life cycle stages observed (Fig. 3).

**Fig. 11.** Example spectra and chromatograms of fragment ions for two peptides corresponding to
a *P. antarctica* flavodoxin identified from the Ross Sea metaproteome sample (peptide sequences
found within Database #1, 1871, contig_31444_1_606_+, 1374 contig_202625_47_661_+; and,
Database #2 contig_175060_39_653_+). Peptide fragmentation spectra are shown in (a) and (c)
and example chromatograms of MS1 intensities as well as with +1 and +2 mass addition for
isotopic distributions is shown (b) and (d), demonstrating the utility of these iron stress biomarkers
in field samples.


**Figure 1.**
(a) 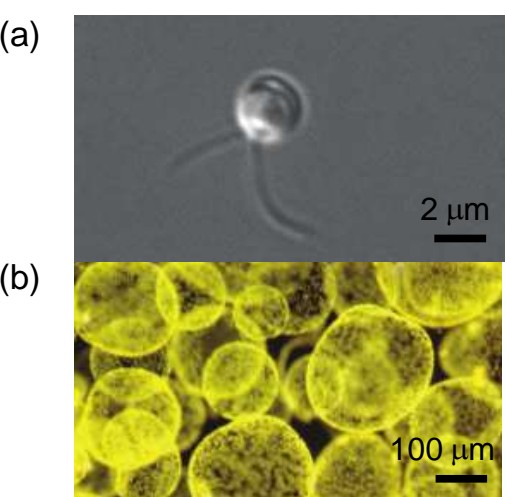

(b)

**Figure 2.**

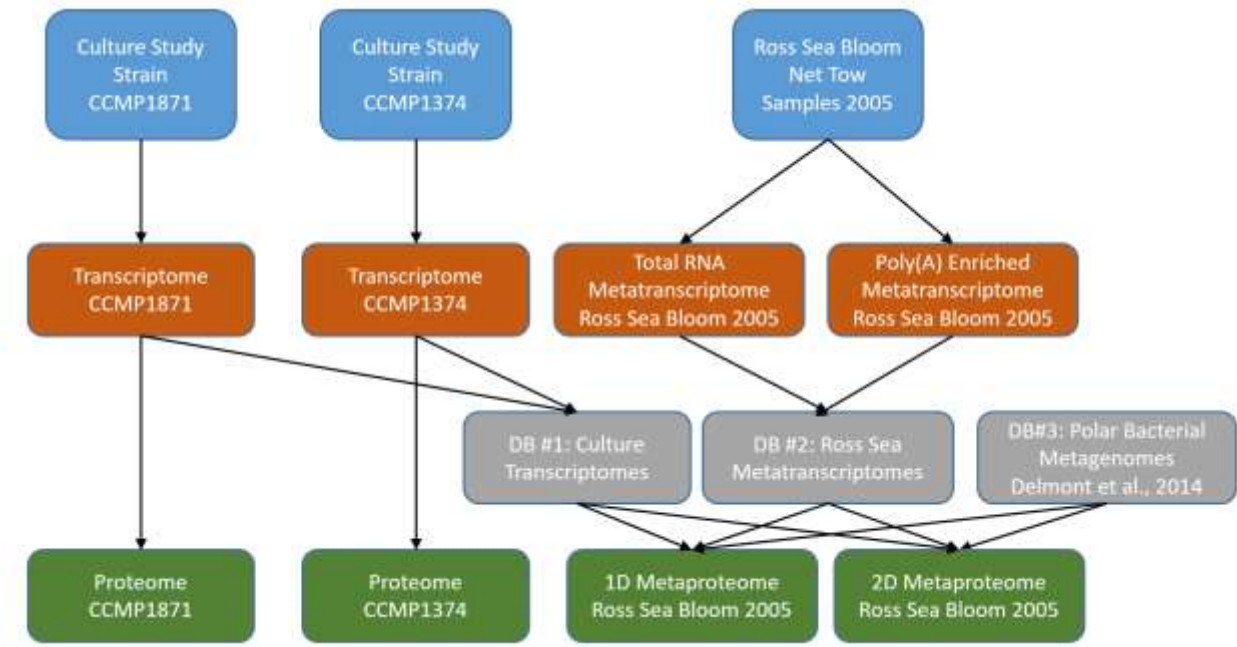


     **Figure 3.**

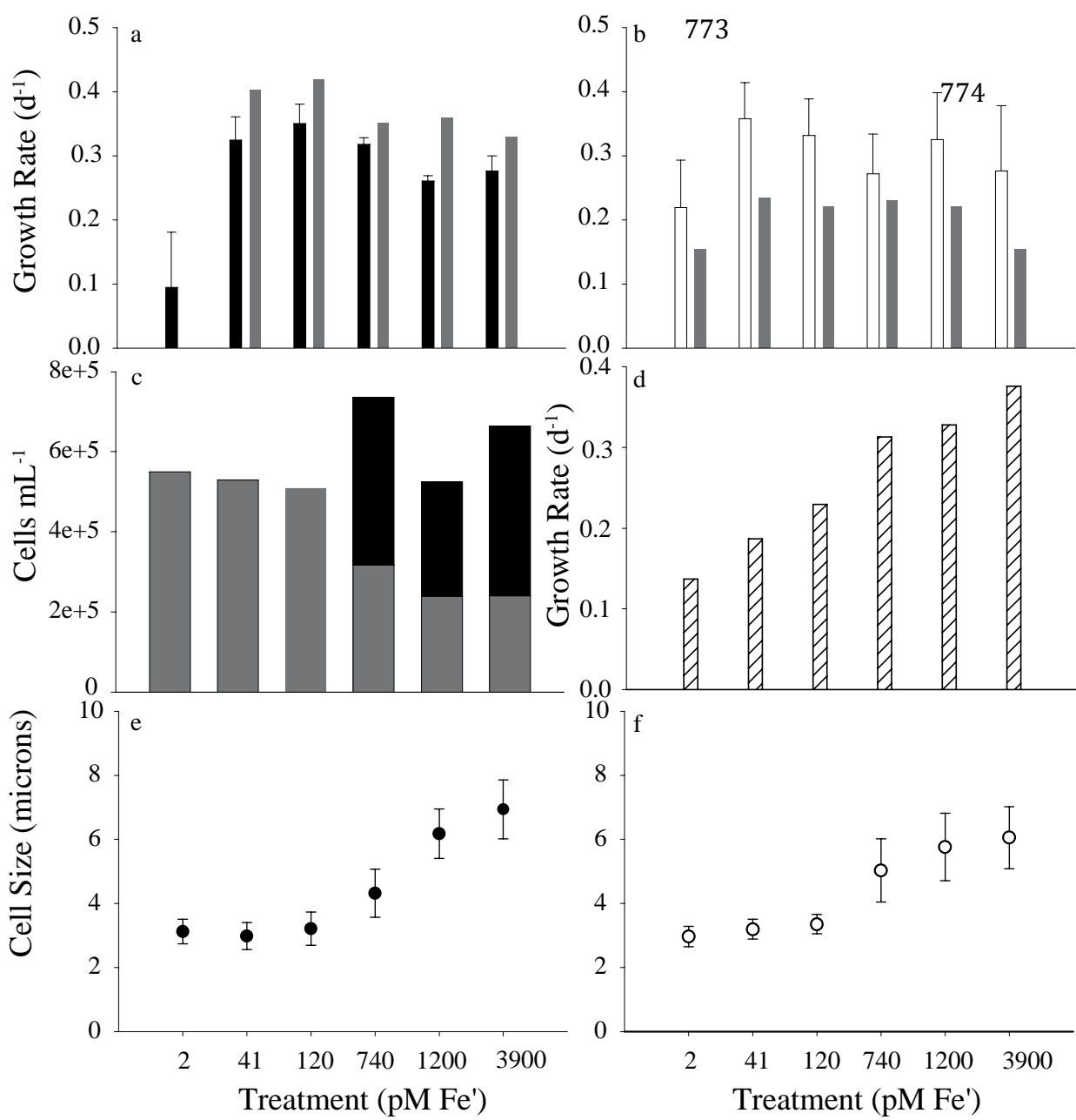

**Figure 4.**

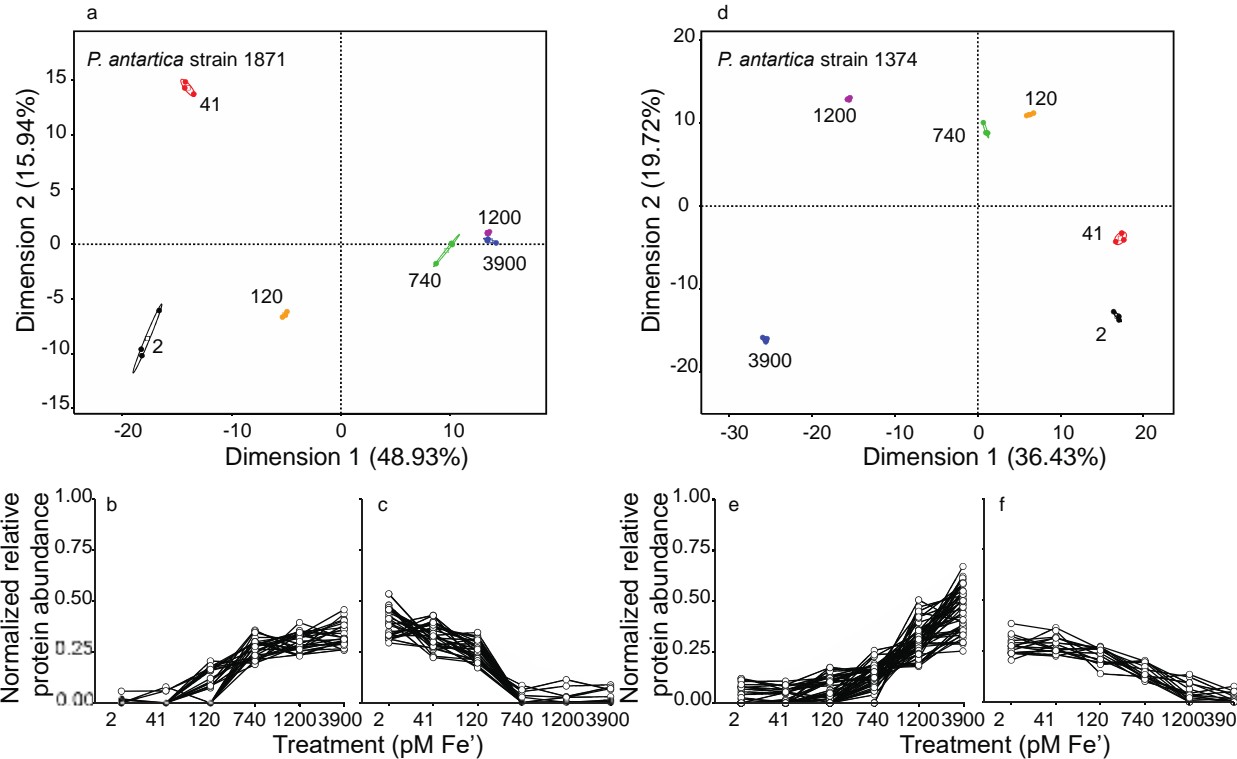

**Figure 5.**


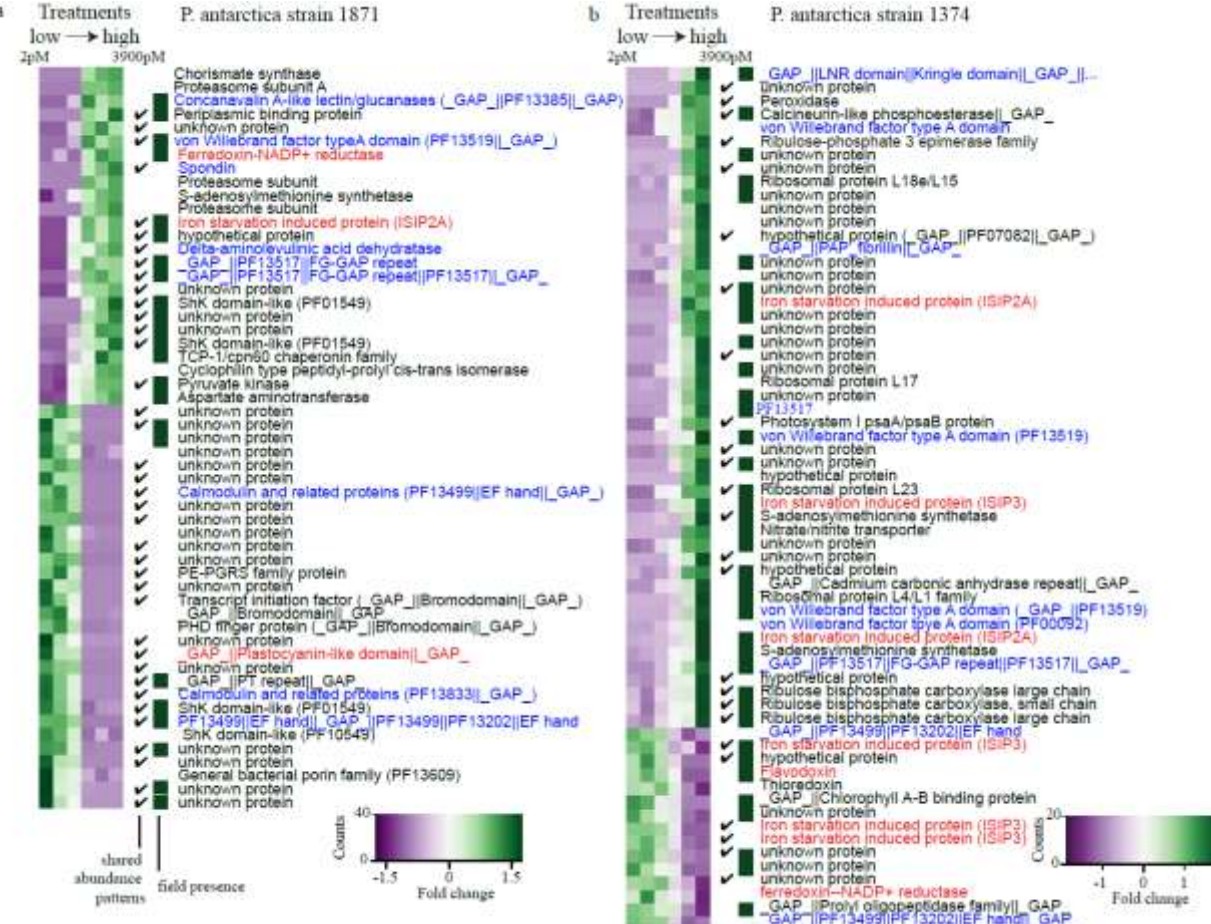


**Figure 6.**

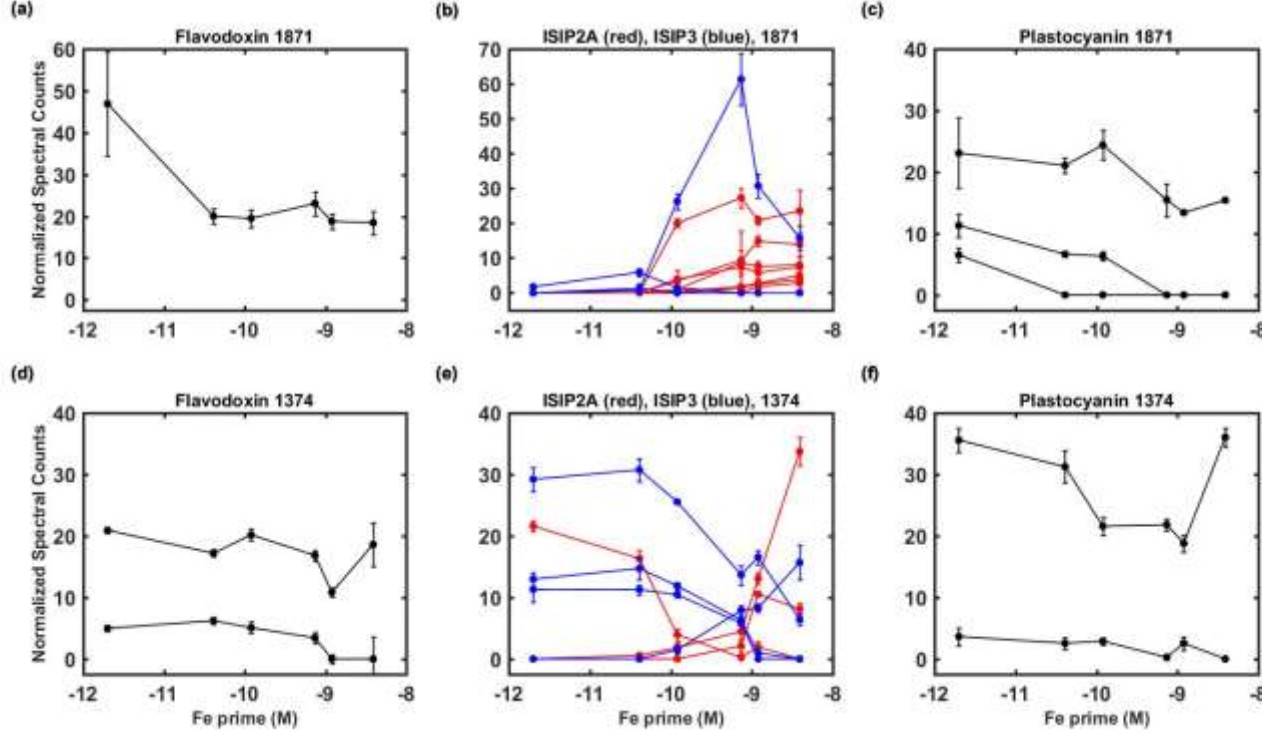

**Figure 7.**

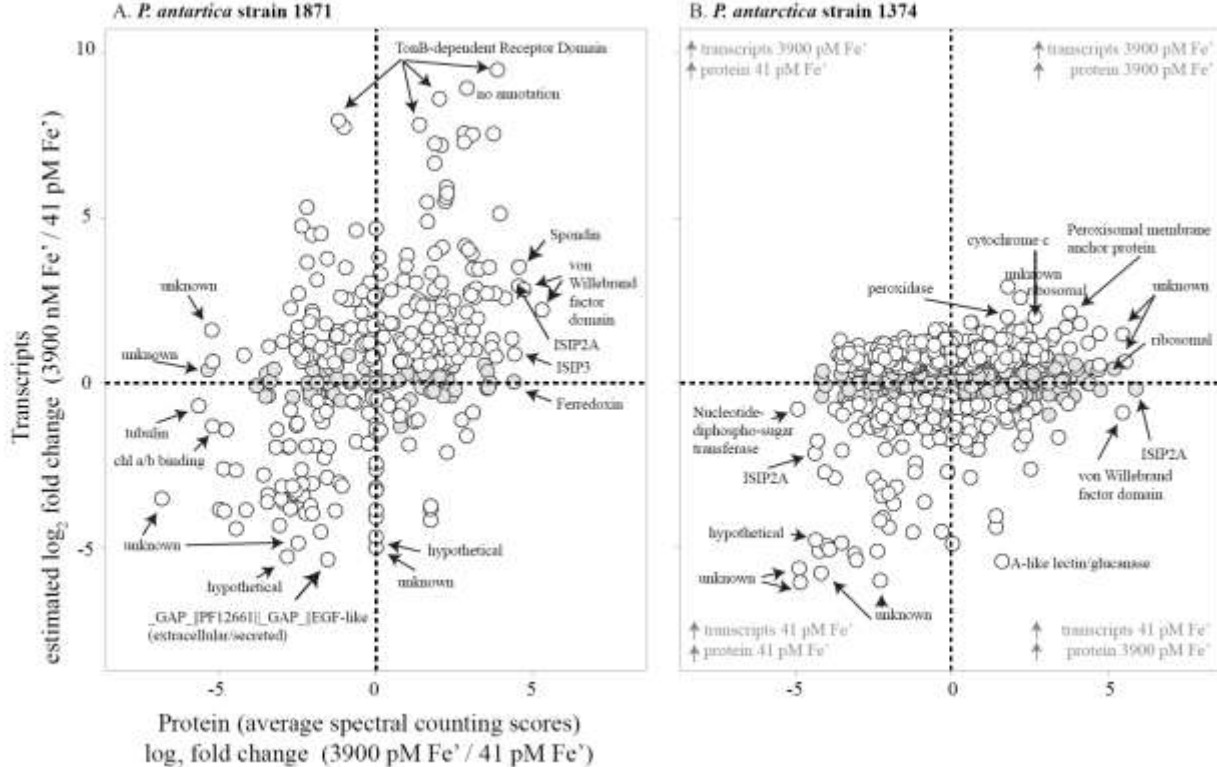


**Figure 8.**



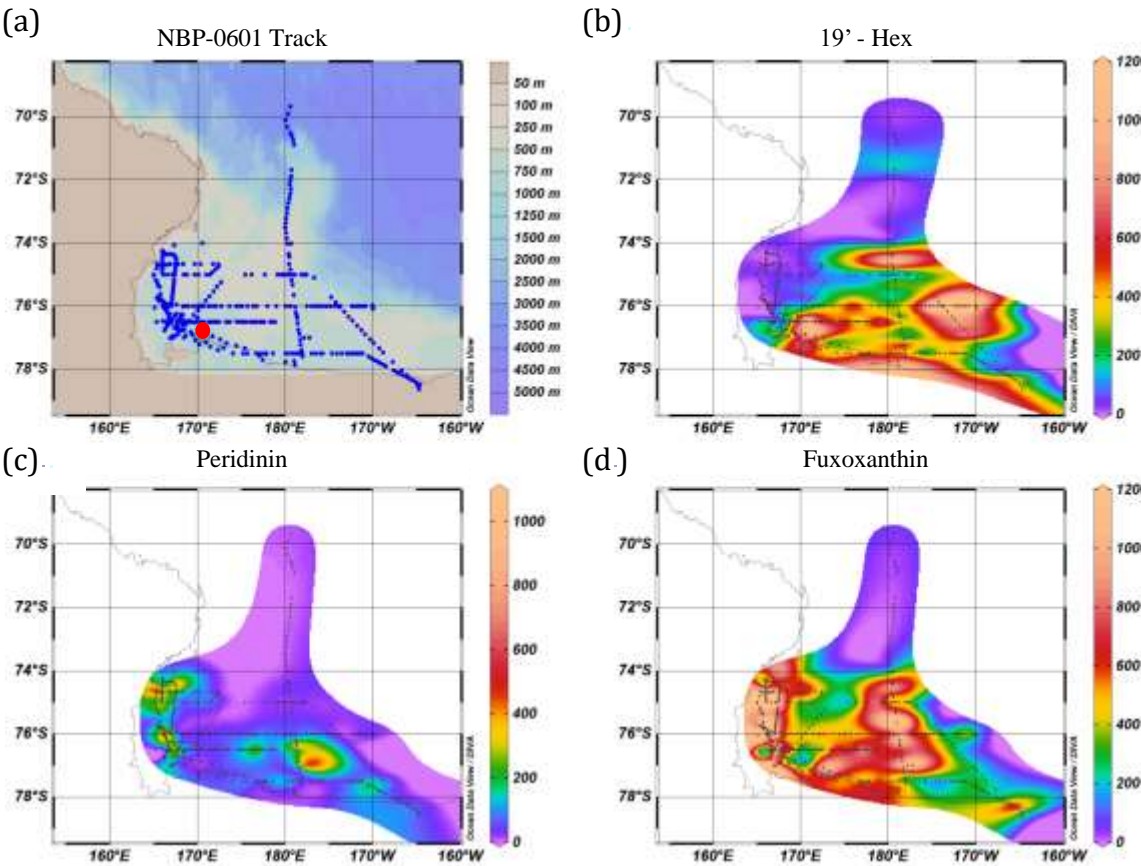

**Figure 9.**

(a)     Unique Peptides by Database          (b)     Genes Identified by Metatranscriptome

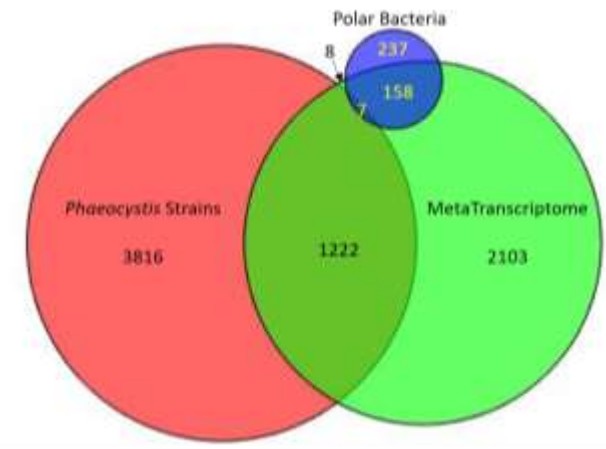

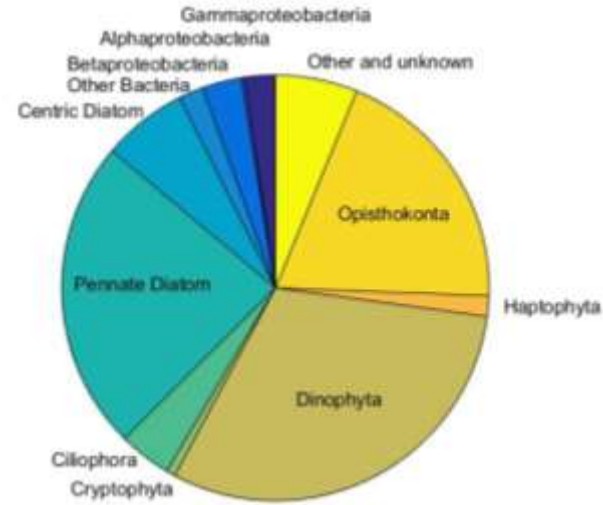


(c)     Proteins by Bacterial Metagenome     (d)     Proteins Identified by Metatranscriptome

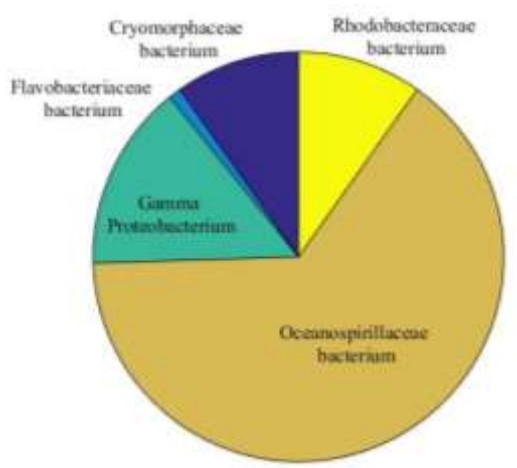

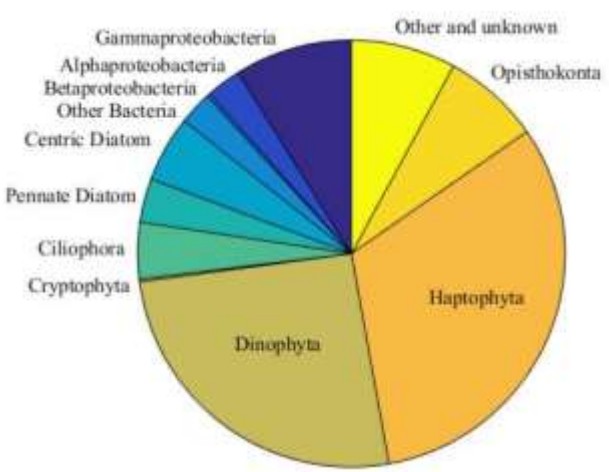

**Figure 10.**

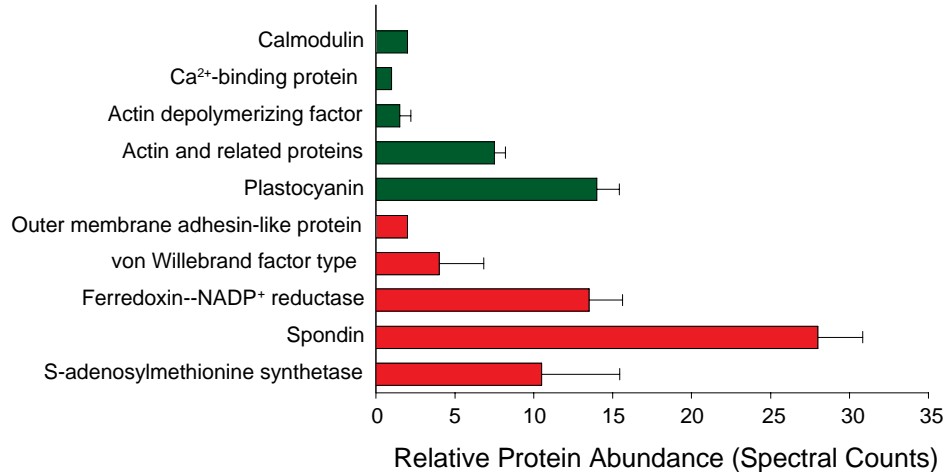

**Figure 11.**

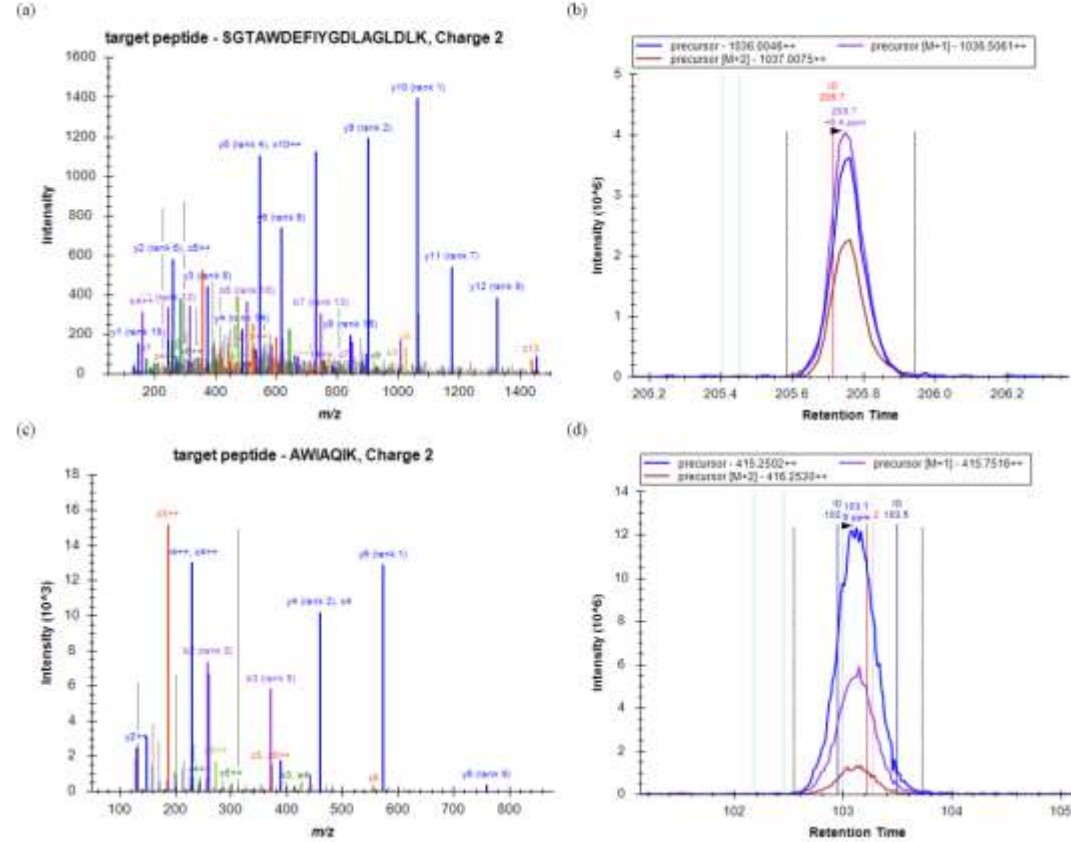

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
