# Peer review of "Colony formation in *Phaeocystis antarctica*: connecting molecular"

_Biogeosciences, 2017_

## Referee Comment (RC1) · Anonymous Referee #1 · 28 Feb 2018

The manuscript addresses the important question whether iron triggers colony formation in Phaeocystis antarctica. Visual observations of batch cultures and an in situ Southern Ocean P. antarctica bloom are linked to (meta)transcriptomic and (meta)proteomic approaches to decipher the underlying processes involved in acclimation to different iron concentrations and colony formation. Considering that P. antarctica is a key player in the global carbon and sulfur cycling, a deeper understanding of its physiology is of highest importance. The authors raised extensive gene/protein expression data for the first time for P. antarctica. I have, however, several concerns with the work presented: - Biological replication in the physiological experiments is lacking (and in the field work, n=2). As far as I understood, each P. antarctica strain was

grown under different iron conditions without replication. Correspondingly, results and discussion rely on single points (but at least statistics was done in a dose-response manner). - Colony formation observations were only taken in case of a single strain 1871, but not for 1374. As this information is lacking, the discussion about the role of iron in colony formation in the latter strain is invalid and, due to a lack of any replication, is not incredibly strongly supported even in the former. This seems to me a weak basis for building the whole paper on the aspect of colony formation. - The second half of the manuscript is a detailed description of metaproteomic data from a P. antarctica bloom with barely a bearing on colony formation, and so the title is not representative for the data presented. Also, this section is very descriptive, and hardly linked with the rest of the manuscript. - I also missed the cell density information for strain 1374 and some more observations on the physiology would have been useful to discuss the data such as at least Fv/Fm. - There was no information about applied cut-off for annotation stringency or detection of differentially expressed genes/proteins in the material and methods section. These facts lead me to ask for major revisions. However, as the combined approach of metatranscriptomics/metaproteomics is very valuable, the manuscript could be rewritten in a shorter form focusing on the physiology of the key player(s) of the Ross Sea bloom. The idea to use laboratory experiments to aid interpretation of field transcriptomes/proteomes is also interesting but not really fleshed out in the current manuscript. Since the point on colony formation as a response to iron supply is very weakly supported by the observations presented and only relates to small part of the data presented, the authors might consider to focus more on cellular iron responses in general.

---

## Author Comment (AC1) · 29 Mar 2018

We thank Reviewer #1 for their comments. While the reviewer appreciates the useful data presented in this study, the reviewer is concerned about the evidence to support colony formation in Phaeocystis antarctica in response to iron. While we could de-emphasize the notion of this connection and shift focus to overall effects of iron, we were surprised by this comment as we feel there are multiple lines of evidence to support this observation. These include (in order of occurrence): 1) visual (anecdotal) observations of clear differences in colony formation in iron treatments that led to the decision to start measuring colonies (Fig 1a). The first strain became clumpy instead

of colonial, having apparently lost the ability to complete its colonies, while the second strain made colonies easily discernible by eye in higher iron treatments. While we acknowledge this is anecdotal, it is worth noting how rare it is to have experiments that produce such strong results that they can be clearly observed with the naked eye, resulting in an on-the-fly change in data collection, initiating the cell type microscopy counts. 2) Cell counts in the treatments with the three highest iron treatments had a majority of cells in the colony form (these could be averaged and standard deviation calculated for above and below iron threshold treatments as an additional sentence if desired/appropriate). This change from non-detectable colonial cells to a majority was a very large difference between low and high iron treatments. 3) Increases in cell size across this gradient observed in both Phaeocystis strains, consistent with the previously documented larger cell sizes of colony cells (Fig 2e). 4) Numerous protein concentration changes that were consistent with a major differences in ultra-structure proteins across the iron gradient (Supplemental Figure 1; Figure Fig 3b,c,e,f; Fig 4). Specifically, the low iron/flagellate cells of both strains clearly have increased abundance of alpha and beta actin and tubulin proteins that are known to be the major proteins within the flagella and haptonema ultrastructure that occur in flagellated cells but not colonial cells (Figure S1; Figure 1). Similarly, the high iron/colonial cells have clear increased abundances of numerous glycoproteins identified (lectins and von Willebrand proteins) known to be involved in the intercellular colony 'skin' in other colonial organisms such as Volvox. The observation of both types of proteins in low/high iron treatments and in both strains is consistent with the clumpy strain being unable to fully complete colony formation likely due loss of some component while in maintenance culture over the years. While we are careful to not exclude the possibility of other types of environmental triggers that could create colonies, in this study we found these four lines of evidence to be convincing and can endeavor to further articulate them in the revision. In addition, recently conducted experiments with strain 1871 in our laboratory have produced similar and consistent results regarding colony formation at varying iron availability.

The reviewer felt that the connection of the metaproteome to the culture study was not clear. We can work to make this clearer in the revision as well. Figure 9 was intended to make this connection, where proteins described earlier in the culture aspects of the study to be associated with the flagellated (green bars) or colonial (red bars) were observed to both be present in the field Phaeocystis net tow metaproteome. The interpretation here is that in order to have an actively growing bloom, both of these diploid cell types are expected to be present and that is consistent with our metaproteome observations of proteins corresponding to each being observed. We acknowledge that there are some methodological aspects of this metaproteome analysis that add some length to this study, but given that metaproteome analysis of eukaryotic algal populations in field samples is relatively new, we feel strongly that it is quite important to have some of this methodological discussion (about database types) included in the manuscript to allow transparency about methodological challenges and successes, and to enable future studies to build on this. We also agree that there are a small number of samples in this metaproteome, but when the samples were collected in 2006, metaproteomic studies were new and studies at that time had few samples included. We recently acquired a large number of samples from this region from a new expedition, and hence we will be able to build on this culture and small metaproteome analysis to interpret large scale temporal and spatial variability dynamics of natural Phaeocystis antarctica populations in the future.

We thank the reviewer for their constructive comments, and we look forward to revising and incorporating their suggestions into this manuscript.

---

## Referee Comment (RC2) · Anonymous Referee #2 · 8 Jun 2018

**1. OVERALL MANUSCRIPT REVIEW AND MINOR CORRECTION REVISION:**

Bender et al. report the effect of iron on colony formation in P. antarctica through a series of "-omics" scale measurements in the laboratory and directly from the field (in the Ross Sea). As the authors mention, this area of research is garnering increasing interest among a multidisciplinary audience- and rightfully so. The work is a discovery-driven report describing proteome/transcriptome dynamics in response to iron in P. antarctica. The authors have thoroughly described their approaches in the Supporting Information and should be commended for their transparency in data processing, as well as making their data available in multiple public repositories.

I DO recommend acceptance of this manuscript with MINOR revisions, with 3 reservations that I hope the authors consider:

A. The volume of work could greatly benefit from a visual scheme (or diagram or flow chart) in the Supplementary Information section (at a minimum). This will help the reader discern laboratory measurements vs. those collected from the field, while also enabling the depiction of the experimental replication scheme (for both transcriptomics experiments and LC-MS/MS experimental replicates). While the presentation of the data are in order as written, it was at times difficult for me to discern what experiments were conducted where, and how those specific measurements were being placed into a potential mechanism.

As presented, the work encompasses a very large undertaking, spanning approximately 15 years. Samples were collected in the Ross Sea in December 2005, mass spectrometry-based proteome measurements (LC-MS/MS) were acquired in late 2013 to early 2014, and transcriptomic/proteomic data analysis performed between 2014 to the time of manuscript submission in late 2017. One figure to show the body of work-particularly in a Supplementary Figure- would do the amount of work reported here justice.

B. Please "soften" some of the semantics in the discussion of the results. There are many instances itemized below that I think could help the audience interpret the measurements more accurately—particularly the proteome measurements.

For instance, line 31 in the Abstract: "...327 and 436 proteins significantly different between low and high iron strains..." Analytically, this is just what was able to be measured in the complex matrix of the samples in question, so perhaps rephrasing using the specific words "detected" or "measured" are more accurate.

Reporting the results to a multidisciplinary audience is challenging and many different fields report measurements differently, but presenting the measurements in context to what they are could- in general across the paper- assist with what the authors are truly

describing as support for a more focused hypothesis of a mechanism (which would be presented in a subsequent paper, for example, as this work is very much discovery driven).

C. A citation should be added on page 21 (lines 476-477) where the reference to "...prokaryotic components of natural communities..." is cited, first by Ram et al. 2005. An Analytical Chemistry publication in early 2005 demonstrated the proof-of-principle that was used by Ram et al. in a significantly more complex sample matrix consisting of both eukaryotic and prokaryotic proteins.

VerBerkmoes NC, Hervey WJ, Shah M, Land M, Hauser L, Larimer FW, Van Berkel GJ, Goeringer DE. Anal. Chem., 2005, 77 (3), pp 923–932. DOI: 10.1021/ac049127n https://pubs.acs.org/doi/abs/10.1021/ac049127n

The paper above showed that commercial mass spectrometry instrumentation and software were capable of measuring complex metaproteomes and reported the limit of detection for E. coli across a range of protein extracts from multiple model organisms (Arabidopsis, the yeast, and at least 3 other microbes). This work also presented peptide-spectrum matching (as used in Bender et al.) in a complex system.

2. SUGGESTED MINOR REVISIONS FOR CORRECTION:

Line 34" "...dynamic in range..." - is this referring to dynamic range of plastocyanin or the complexity of the sample matrix (analytically). I think that this could be worded more clearly for the reader.

Line 40: EF domains - the acronym for "EF" was not presented in multiple descriptions of these proteins- that needs to be made clearly to the reader.

Line 70: One of several instances of a spaced needing to be inserted between the end of a sentence and/or a reference.

Line 149: Microsoft Office's "auto-incorrect" feature changed the abbreviation for acetonitrile (ACN) to "CAN." The abbreviation should be changed to ACN, MeCN, or another abbreviation of the organic content in solvent B.

Lines 156-157: "Normalized spectral counts were generated from Scaffold. . ."

Line 170: please make the word "transcriptome" plural; the methods describe processing 6 of them.

Line 176: Please include the version of CLCbio (and the vendor, Qiagen) in the Supporting Information.

Line 177: Please see comment above about line 70; a space needs to be inserted after "FragGeneScan" and the reference.

Line 213: Please see comments above about line 70 & 177: a space is needed between ". . .(Thermo Scientific). . ." and the reference to Eng 1994.

Lines 268-269: Please refer to the semantics (above) on how ". . .436 proteins were identified as significantly different in relative protein abundance between "low" and "high" Fe." In my opinion, they were detected or observed at different iron concentrations and the amount of iron MAY modulate expression of these proteins.

Line 281: please spell out what the EF-hand domain is again; please see comment about line 40.

Lines 282-285: How many of the proteins of unknown function were previously listed as hypothetical proteins (e.g. potential protein-coding ORFs that have not yet been identified at the protein level)? That number is also a potentially interesting result from the proteome measurements.

Lines 328-330: With respect to drawing a hypothesis from the measurements, it appears that the word "expressed" is missing before constitutive(ly)- unless I misunderstand the conclusions that are being drawn.

Line 347: Please insert the word "potentially" before "related" - this inference is based upon sequence homology for functional inference

Line 383: Please include the word "potentially" before "contribute" - the ECM-related adhesion of cell (potential function of the proteins detected) is based upon sequence homology for functional inference

Line 409: Please see the previous comments above on semantics about the verb "produced" - the similar protein families that are being referred to were measured (and then filtered, etc., etc.) before making this conclusion.

Line 419: Please add a space before the word "Additionally..." - the reference preceding this word needs a space.

Line 420: Please spell out the "EF" domain proteins that are being described here. By this point in the paper, I am presuming that this is the canonical domain that is in calmodulin; but, are the proteins that were measured EF-containing or EF-like, etc. Moving this up in the Abstract for the reader would make things a bit clearer.

Lines 451-452: Semantics of "relative newness" - perhaps novelty of omics application(s)? Relatively new maybe? Or perhaps not include the word "relative" at all, since spectral abundance measurements are included and discussed in depth.

Lines 476-477: Please see the request to cite VerBerkmoes et al. Analytical Chemistry, 2005, as this reference provided a benchmark (proof-of-principle) for the Ram et al. paper presented- in a more complex biological system at that. The Analytical Chemistry reference also shows that detection of eukaryotic proteins in a complex matrix by peptide-spectrum matching was plausible on commercial instruments and software ∼13 years ago.

Line 517: there is an inadvertent space between "colonies ." that needs to be deleted.

Lines 592-593: Could this sentence please be re-worded for clarity? Again, this goes to semantics, in my opinion- perhaps the measurements "Yield preliminary insight into structural remodeling process(es)..." going into line 594. As written, the "provides a first window into the complex..." could be rephrased to help the reader know what is

being emphasized from the measurements.

Line 613 (Acknowledgements): Please insert a space to the person that the authors are indebted to & the NSF award number.

Line 629 (Table 1 legend): As written, the table appears to include a number of differentially expressed [modulated] transcripts; however, there are 3 columns in Table 1 and the number of transcripts appear to not be listed. I am not certain if this is a conversion error (from Word into a PDF, etc.), but the way that I am reading it it appears that values are missing for differentially modulated transcripts.

Line 663 (Figure 3 legend): "...PCA of the full proteomes for each condition..." - I am not able to discern if "full proteome" refers to the proteome profile that has been measured, all of the entries in the database(s) used, or something else. I believe that what is being listed is a PCA plot of the proteins measured, but "full proteomes" is semantically misleading.

Line 729: Please capitalize "ms1" as written in the Figure 10 legend. Given the amount of time that went into making these figures merits this small change.

Supplementary Information: For a multi-omits study, the Supplementary Information provide adequate information for a team of researchers to determine how and what were measured. However, as strenuously suggested above, a visual depiction or representation of the ALL of the measurements performed would greatly benefit the reader- especially since this work spans 2 omics measurements, multiple replicates, and BOTH samples measured from the lab & field.

With respect to extensive description of peptide-spectral matching in metaproteomes in the Supplementary Information (SI, pages 11-12), the authors have taken considerable time justifying an approach that is commonplace among the studies of microbiomes- particularly those from limited environmental origin. While novel applications are in active development to improve the state of proteome informatics, the

cross-correlation scores (and appropriate delta CN values) listed by the authors have been used extensively over the past 16 years (Tabb et al. J. Proteome Res. 2002. https://pubs.acs.org/doi/abs/10.1021/pr015504q ) and the justification, as written, for the peptide-spectrum matching approach is widely used in metaproteomics.

---

## Author Comment (AC2) · 9 Jun 2018

We thank the reviewer for their detailed comments on the manuscript. We appreciate the positive comments regarding transparency and submission to data repositories. With regards to the minor revisions on three points and minor edits, these suggestions will be undertaken and incorporated into the revised manuscript. The suggestion of a workflow figure for the supplemental material is a particularly useful one to help document the different datasets and their interrelation, particularly in the meta-omic connections. The suggestion to soften of language with regards to limiting discussion to those proteins detected is a useful clarification and reminder of the depth of our current

analyses. And we also appreciate the suggested reference and will include that. We also thank the reviewer for the detailed specific edits suggested for the manuscript and will incorporate them. We were not aware of the origin of the EF-hand motif descriptor, but have since learned it is related to E and F helices in the original protein structure where it observed. We do not foresee any issues with incorporating these changes in a timely manner.

---

## Author Response (AR1)

**Minor revisions for** bg-2017-558

We are very appreciative of the reviewers' and editor's constructive suggestions and have followed all suggestions. The changes to the manuscript text were relatively minor (small edits or changing phrases for clarity or softening of message based on reviewer comments). Below is a summary of changes made the manuscript.

**Reviewer #1:**

The reviewer's concerns regarding colony formation were addressed in the response to reviewer comment. Specific changes resulting from the reviewer's comments are:

Given the reviewer's skepticism we have added an additional experiment that showed the same trends to the supplemental materials and this sentence was added to the manuscript discussion: "This influence of iron on colony abundance was observed in an additional experiment, where colonial cells were again absent at the lowest three iron concentrations and were present at the three higher concentrations (Fig. S10)." This new Fig S10 is added at the end of this document as well.

A clause was added to describe the ability to observe colonies visually in the high iron treatments, and on subsequent experiments. The title was changed (see editor's request below).

Information regarding the cut-off stringency was included in the supplemental materials.

**Reviewer #2:**

All changes requested by reviewer #2 were made to the manuscript and supplemental materials with the minor exception of the semantics of the verb "produced" comment, which we were unsure what the reviewer intended. We tried to clarify this sentence as well.

The mistake in Table 1 legend that described RNA data was removed.

The suggested workflow figure was added as a new Figure 2:

[Figure]

**Associate Editor's report:**

The title was modified as requested to "Colony formation in *Phaeocystis antarctica*: connecting molecular mechanisms with iron biogeochemistry". Description of the iron-induced colony formation observed in this study was checked for any overstatements and/or softened.

The Luxum et al. reference was added.

Changes were made to state that the Ross sea is *one* of the most productive regions as requested in the editor's initial review: (Editor's comments: P. 4 Line 86-88: Not really correct: " The Ross Sea is one of the most productive regions". Check the productivity values in Arrigo (2008), and Arrigo et al. (2015; J. Geophys. Res. Oceans, 120, 5545–5565, doi:10.1002/2015JC010888.). Lovenduski and Sarmiento do not refer to the Ross Sea but the Southern Ocean in general.)

**Additional changes in revision:**

Figure 6 lines and symbols made darker for improved readability.

In the intervening time during since submission, an additional re-analysis of the metaproteome Ross Sea sample was completed. These were the two Ross Sea 2005 samples presented in the original submission, but now also analyzed by 2-dimensional chromatography (8 hour runs) in addition to the 1-dimensional chromatography runs (3h). The resulting dataset has significantly more peptide and protein identifications (an increase in peptides from 2013 to 3816 using database #1 for example). While not removing any original data from the original submission, we have now supplemented this minor revision version with this deeper analysis, keeping the original reviewed 1D dataset for comparison in the text and supplemental materials. Notably, the increased depth of the proteome in the 2D additional data *does not alter the conclusions of the manuscript at all*, but will serve to provide readers greater supplemental information to explore the metabolism of the region more fully and allow the development of future mass spectrometry based targeted assays by having nearly double the amount of discovered peptides that are being reported in the supplemental materials.

Specific changes that were made by the addition of the 2D data are listed here:

- Additional values added to the metaproteome discussion providing both 1D and 2D values. (format: "### (in 2D; ### in 1D)".
- Additional filled boxes added to Figure 5 the # of field proteins went from 50 to 61.
- Additional filled boxes added to Supp Figure 1, the # of field proteins went from 26 to 30.
- Figure 9 a) peptide numbers in Venn diagram updated to 2D values, with trends are very similar despite ~2-fold increase in peptide identifications. b) no change (RNA data), c) updated to 2D data, slight expansion of *Oceanospirillaceae* (not this database search produced few hits in either 1D or 2D likely due to minimal bacterial recovery likely caused by the use of a coarse net. The text has been updated to include this methodological interpretation.), d) updated to 2D data, very similar trend.
- Additional metaproteome statistics added to Table 2
- Methods paragraph added to supplemental:

  Samples were analyzed with a Thermo Fusion mass spectrometer following online 2-dimension active modulation liquid chromatography using a Dionex Ultimate3000 RSLCnano system with an additional RSLCnano pump. The first column separation utilized a nonlinear 8 hour pH = 10 gradient (10 mM Ammonium Formate and 10mM ammonium

Formate in 90% acetonitrile) on a PLRP-S column (200 μm x 150 mm, 3 μm bead size, 300Å pore size, NanoLCMS Solutions). The eluent was diluted inline (10 μL/min 0.1% Formic acid) then trapped and eluted every 30 min on alternating dual traps (300 μm x 5 mm, 5 μm bead size, 100 Å pore size, C18 PepMap100, Thermo Scientific). The alternating traps were eluted at 500 nL/min onto a C18 column (100 μm x 150 mm, 3 μm particle size, 120 Å pore size, C18 Reprosil-Gold, Dr. Maisch GmbH packed in a New Objective PicoFrit column) with a 30 min nonlinear gradient (0.1% Formic Acid and 0.1% Formic Acid in 99.9% Acetonitrile) on a Thermo Flex ion source attached to the mass spectrometer.

- Supplemental Figure S3 added addition panels B and D for the 2D metaproteome datasets in addition to 1D datasets.
- Supplemental dataset 2 The additional 2D dataset and its peptides found from the three databases were added to as three new tabs, the metadata and summary tab were updated.
- Corresponding additional Venn diagram peptide count data was added to the supplemental dataset 2 (used in the Figure 9a, original 1D data presented as well).

**Figure S10.** An additional experiment on *Phaeocystis antarctica* strain 1871 across six iron treatments using the same approach and methods as that shown in Figure 3 showing the same trend in colony formation at higher iron concentrations. In this experiment, cells were counted at the time of harvest. Note that the number of cells is higher in the lowest three iron treatments due to being given a longer growth period prior to harvest, since biomass is greater in colonial cells due to their larger cell size (see Figure 3e).

[Figure]

---

## Author Response (AR2)

**Authors' response:**

All requested changes were made to the manuscript text document. Thank you for the corrections. Point by point responses are below in italics.

Sincerely,

Mak Saito

**Associate Editor Decision: Publish subject to technical corrections (26 Jul 2018) by Christine Klaas**

Comments to the Author:

Dear Authors,

I am pleased to accept your manuscript for publication in Biogeosciences. Below please find an additional few minor comments for the final version of the manuscript.

Sincerely,

Christine Klaas

Lines 101-103: "Recent in vitro iron addition experiments provide evidence that iron nutrition influences P. antarctica growth in this region, with increasing P. antarctica biomass after iron addition (Bertrand et al., 2007; Feng et al., 2010)."

*done*

Lines 125-126: "continuous light. Each strain was acclimated to growth on one of six growth condition concentrations":

The meaning of this sentence is not clear: was each strain acclimated to one Fe concentration before starting the experiments (which one?), or were both strains acclimated to each one of the six growth conditions before sampling? How long (number of generations) was the acclimation phase?

*Text modified: at least 3 transfers per experiment, >27 generations*

Line 242: "colonies since sexual reproduction is thought to require revisiting the flagellate life cycle stage.": Is sexual reproduction what the authors meant? Please provide a reference for this statement.

*No, sexual reproduction is distinct from colonial/flagellate transitions (and far less understood). This sentence was modified a bit since colonies can either reproduce by flagellate cell production or by budding new colonies. Rousseau et al 1994 added. New text:"* The presence of both colony and flagellate cells is expected in actively growing populations since reproduction can involve returning to the flagellate life cycle stage (Rousseau et al., 1994)."

Lines 304 -306: "6d and 6f, Supplementary Table 1). While during both experiments, cells were growing exponentially at the time of harvest, cell densities of strain 1374 were as much as 7.6 fold higher than in the experiment with strain 1871"

*done*

Line 307: "and as a result the denser 1374 strain might have also experienced iron stress"

*done*

Lines 313-314: "colonial P. antarctica spring bloom in 1998 were consistent with this interpretation, with ferredoxin concentrations below the detection limit and flavodoxin present (Maucher and DiTullio, 2003)."

*done*

Line 328: "and the proteome across the gradient in iron concentrations, we hypothesize that this diversity of iron stress".

*done*

Line 380: "organism, or with the influence of any other environmental stimuli in the genus Phaeocystis"

*done*

Section 3.5, p. 20: For genetic diversity and mixing of P. antarctica populations see: Gäbler-Schwarz and L. K. Medlin and F. Leese (2015). A puzzle with many pieces: the genetic structure and diversity of Phaeocystis antarctica Karsten (Prymnesiophyta). European Journal of Phycology, 50 (1): 112-124. https://doi.org/10.1080/09670262.2014.998295

*New sentence and reference added: Moreover, high genetic diversity has been observed across a large number of P. antarctica isolates and even within isolates co-isolated from a bloom (Gäbler-Schwarz et al., 2015).*

Lines 548-549: " mentioned earlier, division and growth of P. antarctica colonies is believed to require transitioning back through the flagellate life cycle stage": please provide a reference for this statement.

*Rousseau et al 1994 added*